# Rejection via Learning Density Ratios

**Alexander Soen**[†]
Amazon
The Australian National University
`alexander.soen@anu.edu.au`

**Hisham Husain**[⋆]
`hisham.husain@protonmail.com`

**Philip Schulz**
Amazon
`phschulz@amazon.com`

**Vu Nguyen**
Amazon
`vutngn@amazon.com`

## Abstract

Classification with rejection emerges as a learning paradigm which allows models to abstain from making predictions. The predominant approach is to alter the supervised learning pipeline by augmenting typical loss functions, letting model rejection incur a lower loss than an incorrect prediction. Instead, we propose a different distributional perspective, where we seek to find an idealized data distribution which maximizes a pretrained model's performance. This can be formalized via the optimization of a loss's risk with a $\varphi$-divergence regularization term. Through this idealized distribution, a rejection decision can be made by utilizing the density ratio between this distribution and the data distribution. We focus on the setting where our $\varphi$-divergences are specified by the family of $\alpha$-divergence. Our framework is tested empirically over clean and noisy datasets.

## 1 Introduction

Forcing Machine Learning (ML) models to always make a prediction can lead to costly consequences. Indeed, in real-world domains such as automated driving, product inspection, and medical diagnosis, inaccurate prediction can cause significant real-world harm [16, 29, 53, 50]. To deal with such a dilemma, selective prediction and classification with rejection were proposed to modify the standard supervised learning setting [15, 17, 72]. The idea is to allow for a model to explicitly reject making a prediction whenever the underlying prediction would be either inaccurate and / or uncertain.

In classification, a *confidence-based* approach can be utilized, where a classifier is trained to output a "margin" which is used as a confidence score for rejection [7, 31, 71, 59, 53]. The model rejects whenever this confidence score is lower than an assigned threshold value. A key aspect of these approaches is that they rely on good probability estimates $\Pr(\mathsf{Y} \mid \mathsf{X} = x)$ [60], *i.e.*, being calibrated [71, 53]. While some approaches avoid explicit probability estimation, these are typically restricted to binary classification [7, 31, 47]. Empirically, approaches utilizing confidence scores have shown to outperform other methods, even with simple plugin estimates for probabilities [26, 39, 58].

Another *classifier-rejection* approach aims to simultaneously train a prediction and rejection model in tandem [16, 17, 53]. These approaches are theoretically driven by the construction of surrogate loss functions, but in the multiclass classification case many of these loss functions have been shown to not be suitable [53]. For the multiclass classification setting, one approach connects classification with rejection to cost-sensitive classification [14]. In practice, these classification-rejection approaches require models to be trained from scratch using their specific loss function and architecture — if there

---

(†) Work done while interning at Amazon.
(*) Work done while employed at Amazon.

38th Conference on Neural Information Processing Systems (NeurIPS 2024).

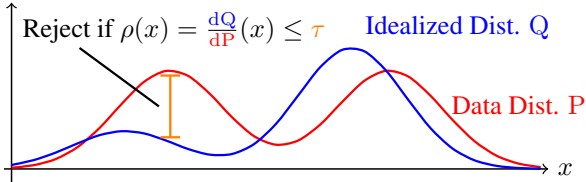

Figure 1: An *idealized distribution* Q is learned to *minimizes* the loss of a model. We then compare Q with the original data distribution P via a density ratio $\rho = \mathrm{dQ}/\mathrm{dP}$. A rejection criteria is defined via threshold value $\tau$.

---

**Algorithm 1** Density-Ratio Rejection

---
**Require:** Model $h$; Divergence $\alpha \geq 1$; Reg. $\lambda > 0$; Threshold $\tau \in [0, 1]$.
  1: Calibrate $h$ if needed.
  2: Calculate density ratio $\rho_\lambda^\alpha$ (either Corollary 4.1 or 4.6).
  3: Normalize $\rho_\lambda^\alpha$ with Monte-Carlo or Eq. (17).
  4: Calculate rejector $r_\tau$ via Def. 3.2.
**output** $r_\tau$

---

is an existing classifier for the dataset, it must be discarded. A recent approach proposes to learn a post-hoc rejector on top of a pretrained classifier via surrogate loss functions [48].

In this work, to learn rejectors we shift from a loss function perspective to a *distributional* perspective. Given a model and a corresponding loss function, we find a distribution where the model and loss *performs "best"* and compare it against the data input distribution to make rejection decisions (see Fig. 1 and Algorithm 1 with equations boxed). As such, the set of rejectors that we propose creates a rejection decision by considering the *density ratio* [67] between a "best" case (idealized) distribution and the data distribution, which can be thresholded by different values $\tau$ to provide different accuracy vs rejection percentage trade-offs. To learn these density ratios for rejection, we consider a risk minimization problem which is regularized by $\varphi$-divergences [1, 19]. We study a particular type of $\varphi$-divergences as our regularizer: the family of $\alpha$-divergences which generalizes the KL-divergence. To this end, one of our core contributions in this work is providing various methods for constructing and approximating idealized distributions, in particular those constructed by $\alpha$-divergences.

The idealized distributions that we consider are connected to adversarial distributions examined in *Distributionally Robust Optimization* (DRO) [63] and the distributions learned in *Generalized Variational Inference* (GVI) [42]; and, as such, the closed formed solutions found for our idealized distribution have utility outside of rejection. Furthermore, when utilizing the KL-divergence and Bayes optimal models, we recover the well known optimal rejection policies, *i.e.*, Chow's rule [15, 72]. Our rejectors are then examined empirically on 6 different datasets and under label noise corruption.

In summary, our contributions are the following:

- We present a new framework for learning with rejection involving the density ratios of idealized distributions which mirrors the distributions learned in DRO and GVI;
- We show that rejection policies learned in our framework can theoretically recover optimal rejection policies, *i.e.*, Chow's rule;
- We derive optimal idealized distributions generated from $\alpha$-divergences;
- We present a set of simplifying assumptions such that our framework can be utilized in practice for post-hoc rejection and verify this empirically.

## 2 Preliminaries

**Notation**    Let $\mathcal{X}$ be a domain of inputs and $\mathcal{Y}$ be output targets. We primarily consider the case where $\mathcal{Y}$ is a finite set of $N$ labels $[N]$, where $[N] \doteq \{0, \dots, N-1\}$. We also consider output domains $\mathcal{Y}'$ which is not necessarily the same as the output data $\mathcal{Y}$, *e.g.*, class probabilities estimates in classification. Denote the underlying (usually unknown) input distribution as $P_{x,y} \in \mathcal{D}(\mathcal{X} \times \mathcal{Y})$. The marginals of a joint distribution are denoted by subscript, *e.g.*, $P_y \in \mathcal{D}(\mathcal{Y})$ for the marginal on the label space. We denote conditional distributions, *e.g.*, $P_{x|y}$. For marginals on $\mathcal{D}(\mathcal{X})$, the subscripting of x is implicit with $P = P_x$. The empirical distributions of P are denoted by $\hat{P}_N$, with $N$ denoting the number of samples used to generate it. Denote the Iverson bracket $[\![p]\!] = 1$ if the predicate $p$ is true and $[\![p]\!] = 0$ otherwise [43]. The maximum $[z]_+ \doteq \max\{0, z\}$ is shorthanded.

**Learning with Rejection**    We first recall the standard *risk minimization* setting for learning. Suppose that we are given a (point-wise) loss function $\ell \colon \mathcal{Y} \times \mathcal{Y}' \to \mathbb{R}_{\geq 0}$ which measures the level of

disagreement between predictions and data. Given a data distribution $P_{x,y}$ and a hypothesis set of models $h \in \mathcal{H}$, we aim to minimize the expected risk w.r.t. $h \colon \mathcal{X} \to \mathcal{Y}'$,

$$\underset{h \in \mathcal{H}}{\operatorname{argmin}} \quad \mathbb{E}_{P_{x,y}}[\ell(Y, h(X))]. \tag{1}$$

One way of viewing learning with rejection is to augment the risk minimization framework by learning an additional rejection function $r \colon \mathcal{X} \to \{0, 1\}$. This can be formally defined using a regularization term $c \in \mathbb{R}_{\geq 0}$ which controls the rate of rejection:

$$\underset{(h,r) \in \mathcal{H} \times \mathcal{R}}{\operatorname{argmin}} \quad \mathbb{E}_{P_{x,y}}[(1 - r(X))\ell(Y, h(X))] + c \cdot P[r(X) = 1], \tag{2}$$

where $\mathcal{R}$ denotes a hypothesis set of rejection functions. Once minimized, a combined model $f$ which can return a rejection token Ⓡ to abstain from predictions can be defined,

$$f(x) = \begin{cases} Ⓡ & \text{if } r(x) = 1 \\ h(x) & \text{if } r(x) = 0. \end{cases} \tag{3}$$

Suppose that $\mathcal{R}$ is complete (contains all possible rejectors). In such a case, one can see that setting $c = \infty$ reduces Eq. (2) to standard risk minimization setting given by Eq. (1). Furthermore, setting $c = 0$ will result in $r$ *always* rejecting, *i.e.*, the case where there is no rejection cost. Some values of $c \in \mathbb{R}_{\geq 0}$ can be redundant. If the loss $\ell$ is bounded above by $B$, then any $c > B$ is equivalent to setting $c = B$ — the optimal $r^\star$ will be to *never* reject. In classification, where $\ell$ is taken to be the zero-one-loss, $c$ is typically restricted to values in $(0, 0.5)$ as otherwise low confidence prediction can be accepted [53] — our work only considers this case. [59] explores the $c \in [0.5, 1]$ scenario.

So far, the learning task has been left general. By considering *Class Probability Estimation* (CPE) [61], we recover familiar optimal rejection and classifier pairs.

**Theorem 2.1** (Optimal CPE Rejection / Chow's Rule). *Let us consider the binary CPE setting, where $\mathcal{Y} = \{0, 1\}$, $\mathcal{Y}' = [0, 1]$, and $\ell$ be any proper[1] loss function [60] (e.g., log loss). Then w.r.t. Eq. (2), the optimal classifier is given by $h^\star(x) = P[Y = 1 \mid X = x]$ and the optimal rejector is given by $r^\star(x) = [\![\mathbb{E}_{Y \sim h^\star(X=x)}[\ell(Y, h^\star(x))] \geq c]\!]$.*

The theorem can easily be generalized to non-binary cases. We note that Theorem 2.1 is a generalization of the well known Chow's rule of classification with rejection [15, 14]. Indeed, taking $\ell$ to be the zero-one-loss function, we get $r^\star(x) = [\![|2 \cdot P[Y = 1 \mid X = x] - 1| \geq 1 - c]\!]$. One can further clarify this by noticing that $\Pr[r(X) = 1] = \mathbb{E}_P[r(X)]$. In the general proper loss case, the optimal rejector is thresholding a generalized entropy function (known as the conditional Bayes risk [60]). This would correspond to thresholding the class probabilities $P[Y = y \mid X = x]$ (via the *Bayes posterior* $h^\star$), but with different thresholding values per class $y \in \mathcal{Y}$ (unless $\ell$ is a symmetric loss function).

Although the objective of Eq. (2) follows the *cost-based model* of rejection [15], other models of rejection exist in the literature. Alternatively, the *bounded improvement model* of rejection [54, 29] maximizes coverage (non-rejection rate) whilst maintaining a constraint on the performance of the rejection measured by the selective risk (the first term of Eq. (2) inversely weighted by the coverage). Optimality conditions of the bounded improvement have been explored in [27]. In the *bounded abstention model*, the constraint and objective is switched – the selective risk is minimized with a constraint on minimum coverage [54]. Our objective function Eq. (2) (and the cost-based model) can be interpreted as a change in the type of risk considered and a change in the hard coverage constraint to a soft constraint w.r.t. bounded abstention. The optimal strategies of these approaches are explored in [28]. Alongside Section 1, further details about rejection can be found in [34].

**Generalized Variational Inference** Generalized Variational Inference (GVI) [42] provides a framework for a generalized set of entropy regularized risk minimization problems. In particular, GVI generalizes Bayes' rule by interpreting the Bayesian posterior as the solution to a minimization problem [73]. The Bayes' rule minimization is given by,

$$\underset{Q \in \mathcal{D}(\Theta)}{\operatorname{argmin}} \quad \mathbb{E}_{\theta \sim Q}\left[ -\sum_{i=1}^{N} \log p(z_i \mid \theta) \right] + \mathrm{KL}(P \parallel Q), \tag{4}$$

where $\theta \sim Q$ denotes a set of model parameters with likelihood function $p(x \mid \theta)$ and $\{z_i\}$ denotes data. Here, $P \in \mathcal{D}(\Theta)$ denotes the prior and the optimal $Q^\star$ denotes the posterior in Bayes' rule.

---

[1] Properness ensures that the true class probability is the minimizer of the loss in expectation.

For GVI, we generalize a number of quantities. For instance, the 'loss' considered can be altered from the log-likelihood $\log p(z \mid \theta)$ to an alternative loss function over samples $\{z_i\}$. The divergence function $\mathrm{KL}(\mathrm{P} \parallel \mathrm{Q})$ can also be altered to change the notion of distributional distance. Furthermore, the set of distributions being minimized $\mathcal{D}(\Theta)$ can also be altered to, *e.g.*, reduce the computational cost of the minimization problem. One can thus alter Eq. (4) to give the following:

$$\underset{\mathrm{Q} \in \mathcal{Q}}{\operatorname{argmin}} \quad L(\mathrm{Q}) + \lambda \cdot D(\mathrm{P} \parallel \mathrm{Q}), \tag{5}$$

where $\lambda > 0$ and $L$, $D$, and $\mathcal{Q}$ denote the generalized loss, divergence, and set of distribution, respectively. Here $\lambda$ seeks to act as a regularization constant which can be tuned.

In our work, we will consider a GVI problem which changes the loss function and divergence to define an entropy regularized risk minimization problem. Our loss function corresponds to the learning setting. For the change in divergence, we consider $\varphi$-divergence [1, 19], which are otherwise referred to as $f$-divergences [62] or the Csiszár divergence [18].

**Definition 2.2.** Let $\varphi \colon \mathbb{R} \to (-\infty, \infty]$ be a convex lower semi-continuous function with $\varphi(1) = 0$ then the corresponding $\varphi$-divergence over non-negative measures for $\mathrm{P}, \mathrm{Q}$ is defined as:

$$D_\varphi(\mathrm{P} \parallel \mathrm{Q}) \doteq \int_{\mathcal{X}} \varphi\left(\frac{\mathrm{dQ}}{\mathrm{dP}}\right) \mathrm{dP}, \quad \text{if } \mathrm{Q} \ll \mathrm{P}; \qquad \text{otherwise}, \quad D_\varphi(\mathrm{P} \parallel \mathrm{Q}) = +\infty. \tag{6}$$

We note that for $\varphi$-divergences, the regularization constant in Eq. (5) can be absorbed into the $\varphi$-divergence generator, *i.e.*, $\lambda \cdot D_\varphi(\mathrm{P} \parallel \mathrm{Q}) = D_{\lambda \cdot \varphi}(\mathrm{P} \parallel \mathrm{Q})$.

**Distributionally Robust Optimization** A related piece of literature is Distributionally Robust Optimization (DRO) [63], where the goal is to find a distribution that maximizes (or minimizes) the expectation of a function from a prescribed *uncertainty* set. Popular candidates for these uncertainty sets include all distributions that are a certain radius away from a fixed distribution by some divergence. $\varphi$-divergences have been used to define such uncertainty set [8, 22, 45]. Given radius $\varepsilon > 0$, define $B_\varepsilon^\varphi(\mathrm{P}) \doteq \{\mathrm{Q} \in \mathcal{D}(\mathcal{X} \times \mathcal{Y}) : D_\varphi(\mathrm{P} \parallel \mathrm{Q}) < \varepsilon\}$. Given a point-wise loss function $\ell \colon \mathcal{Y} \times \mathcal{Y}' \to \mathbb{R}$, DRO alters risk minimization, as per Eq. (1), to solve the following optimization problem:

$$\min_{h \in \mathcal{H}} \max_{\bar{\mathrm{Q}}_{\mathrm{x,y}} \in B_\varepsilon(\mathrm{P})} \quad \mathbb{E}_{\bar{\mathrm{Q}}_{\mathrm{x,y}}} \left[\ell(\mathsf{Y}, h(\mathsf{X}))\right]. \tag{7}$$

The max over $\bar{\mathrm{Q}}$ is typically over the target space $\mathcal{Y}$. That is, $\bar{\mathrm{Q}}(x, y) = \mathrm{Q}(y) \cdot \mathrm{P}(x \mid y)$ and the max is adversarial over the marginal label distribution [74]. Note that converting the $\varepsilon$-ball constraint into a Lagrange multiplier, the inner optimization over Q in Eq. (7) mirrors Eq. (5). The connection between GVI and adversarial robustness has been previously noted [37].

Typically in DRO and related learning settings, the construction of the adversarial distribution defined by the inner maximization problem is implicitly solved. For example, when $\varphi$ is twice differentiable, it has been shown that the inner maximization can be reduced to a variance regularization expression [22, 21]; whereas other choices of divergences such as kernel Maximum Mean Discrepancy (MMD) yields kernel regularization [66] and Integral Probability Metrics (IPM) correspond to general regularizers [36]. Another popular choice is the Wasserstein distance which has shown strong connections to point-wise adversarial robustness [10, 11, 64].

The aforementioned work, however, seek only to find the value of the inner maximization in Eq. (7) without considering the form the optimal adversarial distribution takes.

## 3 Rejection via Idealized Distributions

We propose learning rejection functions $r \colon \mathcal{X} \to \{0, 1\}$ by comparing data distributions P to a learned idealized distribution Q (as per Fig. 1). An idealized distribution (w.r.t. model $h$) is a distribution which when taken as data results in low risk (per Eq. (1)). Q are idealized rather than 'ideal' as they do not solely rely on a model's performance, but are also regularized by their distance from the data distribution P. Formally, we define our idealized distribution via a GVI minimization problem.

**Definition 3.1.** Given a data distribution $\mathrm{P}_{\mathrm{x,y}} \in \mathcal{D}(\mathcal{X} \times \mathcal{Y})$ and a $\varphi$-divergence, an *idealized distribution* $\mathrm{Q} \in \mathcal{D}(\mathcal{X})$ for a fixed model $h$ and loss $\ell$ is a distribution given by

$$\underset{\mathrm{Q} \in \mathcal{D}(\mathcal{X})}{\operatorname{arginf}} \quad L(\mathrm{Q}) + \lambda \cdot D_\varphi(\mathrm{P} \parallel \mathrm{Q}), \tag{8}$$

where $L(\mathrm{Q}) \doteq \mathbb{E}_{\mathrm{Q}}[L'(\mathsf{X})]$ and $L'(x) \doteq \mathbb{E}_{\mathrm{P}_{\mathrm{y|x}}}[\ell(\mathsf{Y}, h(x))]$.

Given the objective of Eq. (8), an idealized distribution Q will have high mass when $L'(x)$ is small and low mass when $L'(x)$ is large. The $\varphi$-divergence regularization term prevents the idealized distributions from collapsing to a point mass. Indeed, without regularization the distance from P, idealized distributions would simply be Dirac deltas at values of $x \in \mathcal{X}$ which minimize $L'(x)$.

With an idealized distribution Q, a rejection can be made via the density ratio [67] w.r.t. P.

> **Definition 3.2.** Given a data distribution P and an idealized distribution $Q \ll P$, the $\tau$-ratio-rejector w.r.t. Q is given by $r_\tau(x) \doteq [\![\rho(x) \leq \tau]\!]$, where $\rho(x) \doteq dQ/dP(x)$.

Definition 3.2 aims to reject inputs where the idealized rejection distribution has lower mass than the original data distribution. Given Definition 3.1 for idealized distribution, small values of $\rho(x)$ corresponds to regions of the input space $\mathcal{X}$ where having lower data probability would decrease the expected risk of the model. Note that we do not reject on regions with high density ratio $\rho$ as $L'(x)$ would be necessarily small or the likelihood of occurrence $P(x)$ would be relatively small. We restrict the value of $\tau$ to $(0, 1]$ to ensure that rejection is focused regions where $L'(x)$ is high with high probability w.r.t. $P(x)$ — further noting that $\tau = 0$ always rejects.

Although Definitions 3.1 and 3.2 suggests that we should learn distributions Q (and P) separately to make our rejection decision, in practice, we can learn the density ratio $dQ/dP$ directly. Indeed, through the definition of Definition 2.2, we have that equivalent minimization problem:

$$\underset{\rho:\ \mathcal{X} \to \mathbb{R}_+}{\operatorname{argmin}} \quad \mathbb{E}_{P_{x,y}} \left[ \rho(X) \cdot \ell(Y, h(X)) + \lambda \cdot \varphi(\rho(X)) \right]; \qquad \text{s.t.} \quad \mathbb{E}_P \left[ \rho(X) \right] = 1. \tag{9}$$

Such an equivalence has been utilized in DRO previously [23, Proof of Theorem 1]. Notice that the learning density ratio $\rho$ in Eq. (9) is analogous to the acceptor $1 - r(x)$ in Eq. (2). Indeed, ignoring the normalization constraint $\mathbb{E}_P \left[ \rho(X) \right] = 1$, by restricting $\rho(x) \in \{0, 1\}$, letting $\lambda = c$, and letting $\varphi(z) = [\![z = 0]\!]$, the objective function of Eq. (9) can be reduced to:

$$\mathbb{E}_{P_{x,y}} \left[ \rho(X) \cdot \ell(Y, h(X)) \right] + c \cdot P \left( \rho(X) = 0 \right). \tag{10}$$

Given the restriction of $\rho$ to binary outputs $\{0, 1\}$ (and Definition 3.2), we have that $r_\tau(x) = 1 - \rho(x)$. As such, Eq. (10) in this setting is equivalent to the minimization of $r$ in Eq. (2) (with $\mathcal{R}$ as all possible functions $\mathcal{X} \to \{0, 1\}$). Through the specific selection of $\varphi$ and restriction of $r$, we have shown that rejection via idealized distributions generalizes the typical learning with rejection objective Eq. (2).

By utilizing form of $\varphi$-divergences, we find the form of the idealized rejection distributions Q and their corresponding density ratio $\rho$ used for rejection.

**Theorem 3.3.** *Given Definition 3.1, the optimal density ratio function $\rho$ of Eq. (9) is of the form,*

$$\rho_\lambda^\varphi(x) = (\varphi')^{-1} \left( \frac{a(x) - L'(x) + b}{\lambda} \right), \tag{11}$$

*where $a(x)$ are Lagrange multipliers to ensure non-negativity $\rho_\lambda^\varphi(\cdot) \geq 0$; and b is a Lagrange multiplier to ensure the constraint $\mathbb{E}_P \left[ \rho_\lambda^\varphi(X) \right] = 1$ is satisfied. Furthermore, the optimal idealized rejection distribution is given by: $Q^\varphi(x) = P(x) \cdot \rho_\lambda^\varphi(x)$.*

Taking $h$ as the Bayes posterior $h^\star$, $L'$ becomes a function of the ground truth posterior $\Pr(Y \mid X = x)$. Hence taking the output $h$ as a neural network plugin estimate of the underlying true posterior $\Pr(Y \mid X = x)$ (see Section 4.3) yields an approach similar to softmax response, *i.e.*, rejection based on the output of $h$ when it outputs softmax probabilities [29]. Theorem 3.3 presents a general approach to generating rejectors from these plugin estimates, as a function of $\ell$ and $\varphi$.

**Connections to GVI and DRO** The formulation and solutions to the optimization of idealized distributions has several connections to GVI and DRO. In contrast to the setting of GVI, Eqs. (4) and (5), the support of the idealized distributions being learned in Definition 3.1 is w.r.t. inputs $\mathcal{X}$ instead of parameters. Furthermore, the inner maximization of the DRO optimization problem, Eq. (7), seeks to solve a similar form of optimization. For idealized distributions, the maximization is switched to minimization and we consider a distribution over inputs $\mathcal{X}$ instead of targets $\mathcal{Y}$. Indeed, notice that the explicit inner optimization of DRO (in Eq. (7)) over Q can be expressed as the following via the Fan's minimax Theorem [25, Theorem 2]:

$$\sup_{\lambda > 0} \ \inf_{Q_\lambda \in \mathcal{D}(\mathcal{X})} \quad -L(Q_\lambda) + \lambda \cdot \left( D_\varphi(Q_\lambda \parallel P) - \varepsilon \right). \tag{12}$$

Notably, the loss $L(\mathrm{Q})$ in Eq. (7) can be simply negated to make the optimization over $\mathrm{Q}$ in Eq. (12) equivalent to DRO Eq. (8) (noting the only requirement for Eq. (7) to Eq. (12) is that $L(\mathrm{Q})$ is a linear functional of $\mathrm{Q}$). This shows that switching the sign of the loss function changes idealized distributions of Definition 3.1 to DRO adversarial distributions. Indeed, through the connection between our idealized distributions and DRO adversarial distributions, the distributions $\mathrm{Q}^\varphi(x)$ will have the same form as the optimal rejection distributions implicitly learned in DRO.

**Corollary 3.4.** *Suppose $\mathrm{Q}^\varphi_\lambda$ denotes the optimal idealized distribution in Theorem 3.3 (switching $L(\mathrm{Q})$ to $-L(\mathrm{Q})$) for a fixed $\lambda > 0$. Further let $\lambda^\star \in \operatorname{arginf}_{\lambda>0}\{-L(\mathrm{Q}^\varphi_\lambda) + \lambda \cdot (D_\varphi(\mathrm{Q}^\varphi_\lambda \parallel \mathrm{P}) - \varepsilon)\}$. Then the optimal adversarial distribution in the inner minimization for DRO (Eq. (7)) is $\mathrm{Q}^\varphi_{\lambda^\star}$.*

As such, the various optimal idealized distributions (w.r.t. Definition 3.1) for rejection we will present in the sequel can be directly used to obtain the optimal adversarial distributions for DRO.

## 4 Learning Idealized Distributions

In the following section, we explore optimal closed-form density ratio rejectors. We first examine the easiest example — the KL-divergence — and then consider the more general $\alpha$-divergences.

### 4.1 KL-Divergence

Let us first consider the KL-divergence [2] for constructing density ratio rejectors via Theorem 3.3.

**Corollary 4.1.** *Let $\varphi(z) = z \log z - z + 1$ and $\lambda > 0$. The optimal density ratio of Definition 3.1 is,*

$$\rho^{\mathrm{KL}}_\lambda(x) = \frac{1}{Z} \cdot \exp\left(\frac{-L'(x)}{\lambda}\right), \qquad \text{where } Z = \mathbb{E}_\mathrm{P}\left[\exp\left(\frac{-L'(x)}{\lambda}\right)\right]. \tag{13}$$

One will notice that $\rho^{\mathrm{KL}}_\lambda$ corresponds to an exponential tilt [24] of $\mathrm{P}$ to yield a *Gibbs distribution* $\mathrm{Q}^{\mathrm{KL}}$. The KL density ratio rejectors are significant in a few ways. First, we obtain a closed-form solution due to the properties of 'log' and complementary slackness, *i.e.*, $a(\cdot) = 0$. Secondly, due to the properties of $\exp$, the normalization term $b$ has a closed form solution given by the typical log-normalizer term of exponential families.

Another notable property of utilizing a KL idealized distribution is that it recovers the previously mentioned optimal rejection policies for classical modelling with rejection via cost penalty.

**Theorem 4.2** (Informal). *Given the CPE setting Theorem 2.1, if $h = h^\star$ is optimal, then there exists a $r^{\mathrm{KL}}_\tau$ which is equivalent to the optimal rejectors in Theorem 2.1.*

Theorem 4.2 states that the KL density rejectors (with correctly specified $\lambda$ and $\tau$) with optimal predictors $h^\star$ recovers the optimal rejectors of the typical rejection setting, *i.e.*, Chow's rule.

Until now, we have implicitly assumed that the true data distribution $\mathrm{P}$ is accessible. In practice, we only have access empirical $\hat{\mathrm{P}}_N$, defining subsequent rejectors $\hat{\rho}_{\lambda,N}$. We show that for the KL rejector, $\hat{\mathrm{P}}_N$ is enough given the following generalization bound.

**Theorem 4.3.** *Assume we have bounded loss $|\ell(\cdot, \cdot)| \le B$ for $B > 0$, $\hat{\mathrm{P}}_N \subset \mathrm{P}$ with h.p., and $\mathcal{T} \subset \operatorname{Supp}(\mathrm{P})$. Suppose $M = |\mathcal{T}| < +\infty$, then with probability $1 - \delta$, we have that*

$$\sup_{x \in \mathcal{T}} \left|\rho^{\mathrm{KL}}_\lambda(x) - \hat{\rho}^{\mathrm{KL}}_{\lambda,N}(x)\right| \le C \cdot \sqrt{\frac{2}{N} \log\left(\frac{2M}{\delta}\right)},$$

*where $C = \exp(B/\lambda)^3 \cdot \sinh(B/\lambda)$.*

Looking at Theorem 4.3, essentially we pay a price in generalization $M$ for each element $x \in \mathcal{T}$ we are testing for rejection. For generalization, it is useful to consider how $N, M$ changes our rate in Theorem 4.3. If we assume that the test set $\mathcal{T}$ is small in comparison to the $N$ samples used to generate empirical distribution $\hat{\mathrm{P}}$, then the $\mathcal{O}(1/\sqrt{N})$ rate will dominate. A concrete example of this case is when $|\mathcal{X}|$ is finite. A less advantaged scenario is when $M \approx N$, yielding $\mathcal{O}(\log(N)/\sqrt{N})$ — the scenario where we test approximately the same number of data points as that used to learn the rejector. This rate will still decrease with $N \to \infty$, although with a $\log N$ price.

## 4.2 Alpha-Divergences

Although the general case of finding idealized rejection distributions for $\varphi$-divergences is difficult, we examine a specific generalization of the KL case, the $\alpha$-divergences [3].

**Definition 4.4.** For $\alpha \in \mathbb{R}$, the $\alpha$-divergence $D_\alpha$ is defined as the $\varphi_\alpha$-divergence, where

$$\varphi_\alpha(z) \doteq \begin{cases} \frac{4}{1-\alpha^2}\left(1 - z^{\frac{1+\alpha}{2}}\right) - \frac{2}{1-\alpha}(z-1) & \text{if } \alpha \neq \pm 1 \\ -\log z + (z-1) & \text{if } \alpha = -1 \\ z \log z - (z-1) & \text{if } \alpha = 1 \end{cases} \cdot$$

We further define $\psi_\alpha$, where $\psi_\alpha(z) = z^{\frac{1-\alpha}{2}}$ when $\alpha \neq 1$ and $\psi_\alpha(z) = \log z$ when $\alpha = 1$.

Note that taking $\alpha = 1$ recover the KL-divergence. The $\alpha$-divergence covers a wide range of divergences including the Pearson $\chi^2$ divergence ($\alpha = 3$). For the density ratio, $\alpha$-divergences with $\alpha \neq 1$ (*i.e.* not KL) can be characterized as the following.

**Theorem 4.5.** *Let $\alpha \neq 1$ and $\lambda > 0$. For $\varphi_\alpha$, the optimal density ratio rejector $\rho_\lambda^\alpha(x)$ is,*

$$\rho_\lambda^\alpha(x) = \psi_\alpha^{-1}\left(\frac{2}{\alpha - 1} \cdot \left(a(x) - \frac{L'(x)}{\lambda} + b\right)^{-1}\right) \tag{14}$$

$a(x)$ *are Lagrange multipliers for positivity; and b is a Lagrange multiplier for normalization.*

One major downside of using general $\varphi$-divergences is that solving the Lagrange multipliers for the idealized rejection distribution is often difficult. Indeed, the "$\log$" and "$\exp$" ensures non-negativity of the idealized distribution when the input data P is in the interior of the simplex; and also provides a convenient normalization calculation. For $\alpha$-divergences, the non-negative Lagrange multipliers $a(\cdot)$ can be directly solved given certain conditions.

---

**Corollary 4.6.** *Suppose $\alpha > 1$ and $\lambda > 0$, then Eq. (14) simplifies to,*

$$\rho_\lambda^\alpha(x) = \left[\left(\frac{\alpha-1}{2} \cdot \left(b - \frac{L'(x)}{\lambda}\right)\right)^{\frac{2}{\alpha-1}}\right]_+, \tag{15}$$

*where we take non-integer powers of negative values as 0.*

---

On the other hand, for $\alpha \leq -1$, $D_\alpha(\mathrm{P} \parallel \mathrm{Q}) = \infty$ whenever Q is on the boundary whenever P is not on the boundary [3, Section 3.4.1]. As such, we can partially simplify Eq. (14) for $\alpha \leq -1$.

**Corollary 4.7.** *Suppose $\alpha \leq -1$, $\lambda > 0$, and P lies in the simplex interior, then $a(\cdot) = 0$ in Eq. (14).*

Both Corollaries 4.6 and 4.7 can provide a unique rejector policy than the KL-divergence variant. Corollary 4.7 can provide a similar effect when $a(x) \neq 0$ for all $x$. Nevertheless, having to determine which inputs $a(x) \neq 0$ and solving these values are difficult in practice. As such, we focus on $\alpha > 0$. If there are values of $L'(x)$ with high risk, the max will flatten these inputs to 0 in Corollary 4.6. However, if the original model $h$ performs well and $L'(x)$ is relatively small for all $x$, then it is possible that the max is not utilized. In such a case, the $\alpha$-divergence rejectors can end up being similar — this follows the fact that (as $\varphi_\alpha''(1) > 0$) locally all $\alpha$-divergences will be similar to the $\chi^2$ / ($\alpha = 3$)-divergence [56, Theorem 7.20]. This ultimately results in $\alpha$-divergences being similar to, *e.g.*, Chow's rule when $h_y(x) \approx \Pr(\mathsf{Y} = y \mid \mathsf{X} = x)$ in classification via Theorem 2.1.

Among the $\alpha > 0$ cases, we examine the $\chi^2$-divergence ($\alpha = 3$) which results in closed form for bounded loss functions and sufficient large regularizing parameter $\lambda$.

**Corollary 4.8.** *Suppose that $\ell(\cdot, \cdot) \leq B$ and $\lambda > \max_x L'(x) - \mathbb{E}_{\mathrm{P}}[L'(x)]$. Then,*

$$\rho_\lambda^{\alpha=3}(x) = 1 + \frac{\mathbb{E}_{\mathrm{P}}[L'(x)] - L'(x)}{\lambda}. \tag{16}$$

The condition on $\lambda$ for Corollary 4.8 is equivalent to rescaling bounded loss functions $\ell$. Indeed, by fixing $\lambda = 1$, we can achieve a similar Theorem with suitable rescaling of $\ell$. Nevertheless, Eq. (16) provides a convenient form to allow for generalization bounds to be established.

**Theorem 4.9.** *Assume we have bounded loss $|\ell(\cdot, \cdot)| \leq B$ for $B > 0$, $\lambda > 2B$, $\hat{P}_N \subset P$ with h.p., and $\mathcal{T} \subset \text{Supp}(P)$. Suppose $M = |\mathcal{T}| < +\infty$, then with probability $1 - \delta$, we have that*

$$\sup_{x \in \mathcal{T}} \left| \rho_\lambda^{\alpha=3}(x) - \hat{\rho}_\lambda^{\alpha=3}(x) \right| \leq \frac{B}{\lambda} \cdot \sqrt{\frac{2}{N} \log\left(\frac{2M}{\delta}\right)}.$$

Notice that Theorem 4.9's sample complexity is equivalent to Theorem 4.3 up to constant multiplication. Hence, the analysis of Theorem 4.3 regarding the scales of $N, M$ hold for Theorem 4.9.

A question pertaining to DRO is what would be the generalization capabilities of the corresponding adversarial distributions $\hat{Q}_{\lambda,N} \doteq \hat{P}_N \cdot \hat{\rho}_{\lambda,N}$ (through Corollary 3.4). On a finite domain, via Theorems 4.3 and 4.9 and a simple triangle inequality, one can immediately bound the total variation $\text{TV}(\hat{Q}_{\lambda,N}, Q_\lambda) \leq \mathcal{O}(1/\sqrt{N})$, see Appendix M for further details.

### 4.3 Practical Rejection

We consider practical concerns for utilizing the KL-or ($\alpha > 1$)-divergence case, Eqs. (13) and (15), for post-hoc rejection. To do so, we need to estimate the loss $L'(x)$ and rejector normalizer $Z$ or $b$.

**Loss**: The former is tricky, we require an estimate to evaluate $L'(x) = \mathbb{E}_{P_{y|x}}[\ell(Y, h(x))]$ over any possible $x \in \mathcal{X}$ to allow us to make a rejection decision. Implicitly, this requires us to have a high quality estimate of $P_{y|x}$. In a general learning setting, this can be difficult to obtain — in fact it is just the overall objective that we are trying to learning, *i.e.*, predicting a target $y$ given an input $x$. However, in the case of CPE (classification), it is not unreasonable to obtain an *calibrated estimate* of $P_{y|x}$ via the classifier $h \colon \mathcal{X} \to \mathcal{D}(\mathcal{Y})$ [55]. In Section 5, we utilize temperature scaling to calibrate the neural networks we learn to provide such an estimate [32]. Hence, we set $L'(x) = \mathbb{E}_{Y \sim h(x)}[\ell(Y, h(x))]$. For proper CPE loss functions, $\mathbb{E}_{Y \sim h(x)}[\ell(Y, h(x))]$ acts as a generalized entropy function. As such, the rejectors Eqs. (13) and (15) act as functions over said generalized entropy functions. It should be noted that simply considering the softmax outputs of neural networks for probability estimation have seen prior success [29, 39]. This is equivalent to taking the 0-1-loss function [15, 35, 28] and taking a plugin estimate of probabilities via the neural network's output. The study of using plugin estimates have also been explored in the model cascade literature [40].

**Normalization**: For the latter, we utilize a sample based estimate (over the dataset used to train the rejector) of $\mathbb{E}_P[\rho_\lambda^\varphi(X)]$ is utilized to solve the normalizers $Z$ and $b$. In the case of the KL-divergence rejector, this is all that is required due to the convenient function form of the Gibbs distribution, *i.e.*, the normalizer $Z$ can be simply estimated by a sample mean. However, for $\alpha > 1$-divergences $b$ needs to be found to determine the normalization. Practically, we find $b$ through bisection search of

$$\mathbb{E}_{\hat{P}}\left[\left(\frac{\alpha - 1}{2} \cdot \left(b - \frac{L'(x)}{\lambda}\right)\right)^{\frac{2}{\alpha - 1}}\right]_+ - 1 = 0. \tag{17}$$

This practically works as the optimization $b$ is over a single dimension. Furthermore, we can have that $b > \min_x \{L'(x)/\lambda\}$. As an upper bound over the possible values of $b$, we utilize a heuristic where we multiple the corresponding maximum of $L'(x)/\lambda$ with a constant.

**Threshold $\tau$**: In addition to learning the density ratio, an additional consideration is how to tune $\tau$ in the rejection decision, Definition 3.2. Given a fixed density estimator $\rho$, change $\tau$ amounts to changing the rate of rejection. We note that this problem is not limited to our density ratio rejectors, where approaches with rejectors $r \in \mathcal{R}$ via a (surrogate) minimization of Eq. (2) may be required multiple rounds of training with different rejection costs $c$ to find an accept rate of rejection. In our case, we have a fixed $\rho$ which allows easy tuning of $\tau$ given a validation dataset, similar to other confidence based rejection approaches, *e.g.*, tuning a threshold for the margin of a classifier [7].

## 5 Experiments

In this section, we evaluate our distributional approach to rejection across a number of datasets. In particular, we consider the standard classification setting with an addition setting where uniform label noise is introduced [4, 30]. For our density ratio rejectors, we evaluate the KL-divergence based

Table 1: Summary of rejection methods over all baselines and datasets targeting 80% coverage. Each cell reports the "accuracy [coverage]" values, **bold** for most accurate, and s.t.d. reported in Appendix.

| | | Base | KL-Rej | ($\alpha$=3)-Rej | PredRej | CSS | DEFER | GCE |
|---|---|---|---|---|---|---|---|---|
| Clean | HAR | 97.38 [100] | 99.93 [81] | **99.93** [82] | 98.86 [85] | 99.58 [81] | 99.44 [80] | 99.31 [82] |
| | Gas Drift | 94.10 [100] | **99.16** [80] | **99.16** [80] | 98.12 [80] | 98.68 [80] | 98.06 [80] | 97.62 [80] |
| | MNIST | 98.55 [100] | 99.93 [87] | 99.93 [88] | 99.18 [74] | **99.95** [83] | 99.93 [80] | 99.85 [80] |
| | CIFAR-10 | 90.20 [100] | **97.22** [80] | 97.17 [80] | 91.40 [74] | 95.45 [81] | 93.72 [81] | 94.25 [81] |
| | OrganMNIST | 89.10 [100] | **96.55** [80] | 96.52 [80] | 93.79 [82] | 94.49 [80] | 93.47 [80] | 93.68 [80] |
| | OctMNIST | 91.93 [100] | 97.08 [81] | **97.18** [80] | 93.43 [86] | 95.40 [81] | 94.66 [85] | 94.91 [87] |
| Noisy (25%) | HAR | 96.51 [100] | 98.56 [81] | 98.56 [81] | 97.22 [80] | 97.82 [81] | 97.78 [69] | **98.85** [80] |
| | Gas Drift | 93.84 [100] | 97.30 [80] | 97.28 [80] | 95.87 [82] | 98.71 [80] | **99.02** [77] | 97.52 [75] |
| | MNIST | 97.88 [100] | 99.89 [80] | 99.89 [81] | 98.00 [93] | 99.94 [80] | 99.93 [81] | **99.95** [81] |
| | CIFAR-10 | 85.31 [100] | 92.25 [82] | **92.50** [81] | 85.84 [88] | 89.58 [82] | 90.93 [80] | 92.22 [81] |
| | OrganMNIST | 89.10 [100] | 95.86 [82] | **96.29** [81] | 93.40 [84] | 96.74 [81] | 94.67 [80] | 94.48 [80] |
| | OctMNIST | 91.89 [100] | **97.17** [80] | 97.10 [81] | 93.42 [83] | 95.49 [81] | 94.08 [89] | 94.63 [78] |

rejector (Corollary 4.1) and ($\alpha$=3)-based rejectors (Corollary 4.6) with 50 equidistant $\tau \in (0, 1]$ values. For our tests, we fix $\lambda = 1$. Throughout our evaluation, we assume that a neural network (NN) model without rejection is accessible for all (applicable) approaches. For our density ratio rejectors, we utilize the log-loss, practical considerations in Section 4.3, and Algorithm 1.[2]

To evaluate our density ratio rejectors and baselines, we compare accuracy and acceptance coverage. Accuracy corresponds to the 0-1 loss in Eq. (1) and the acceptance coverage is the percentage of non-rejections in the test set. Ideally, a rejector smoothly trade-offs between accuracy and acceptance, *i.e.*, a higher accuracy can be achieved by decreasing the acceptance coverage by rejecting more data. We study this trade-off by examining multiple cut-off values $\tau$ and rejection costs $c$.

**Dataset and Baselines** We consider 6 multiclass classification datasets. For tabular datasets, we consider the gas drift dataset [68] and the human activity recognition (HAR) dataset [5]. Each of these datasets consists of 6 classes to predict. Furthermore, we utilize a two hidden layer MLP NN model for these datasets. We consider the MNIST image dataset [46] (10 classes), where we utilize a convolutional NN. Additionally, we consider 3 larger image datasets with ResNet-18 architecture [33]: CIFAR-10 [44] (10 classes); and OrgMNIST / OrganSMNIST (11 classes) and OctMNIST (4 classes) from the MedMNIST collection [69, 70]. These prediction models are trained utilizing the standard logistic / log-loss without rejection and then are calibrated via temperature scaling [32]. For each of these datasets, we utilize both clean and noisy variants. For the noisy variant, we flip the class labels of the train set with a rate of 25%. We note that the test set is clean in both cases. All evaluation uses 5-fold cross validation. All implementation use PyTorch and training was done on a `p3.2xlarge` AWS instance.

We consider 4 different baseline for comparison. Each is trained with 50 equidistant costs $c, \tau \in [0, 0.5)$, except on OctMNIST which uses 10 equidistant costs (selecting $c, \tau$ discussed in Appendix P). One baseline used corresponds to a modification of [49]'s cross-entropy surrogate approach (DEFER) originally used for the learning to defer literature (see [14, Appendix A.2]). This approach treats the rejection option as a separate class and solves a $|\mathcal{Y}| + 1$ classification problem. A generalization of DEFER is considered which utilizes generalized cross-entropy [75] as its surrogate loss (GCE) [13]. We also consider a cost-sensitive classification reduction (CSS) of the classification with rejection problem [14] utilizing the sigmoid loss function. The aforementioned 3 baselines all learn a model with rejection simultaneously, *i.e.*, a pretrained model cannot be utilized. We also consider a two stage predictor-rejector approach (PredRej) which learns a rejector from a pretrained classifier [48].

**Results** Table 1 presents a tabular summary of accuracy and coverage values when targeting 80% coverage values; and Fig. 2 presents a summary plot of the acceptance coverage versus model accuracy after rejection, focused around the 60% to 100% coverage region. This plot is limited to HAR, Gas Drift, and MNIST due to space limitations however the deferred datasets show curves where the density ratio rejector dominate with better accuracy and coverage in the plotted region, with corresponding extended plots for all datasets in Appendix Q . Over all folds for MNIST our density ratio rejectors take approximately $\approx 1/2$ hour to fit. A single baseline (fixed $c$) takes upwards of 2 hour for a single fold. Overall, given that the underlying model is calibrated, we find that our

---

[2]Our rejector's code public at: `https://github.com/alexandersoen/density-ratio-rejection`.

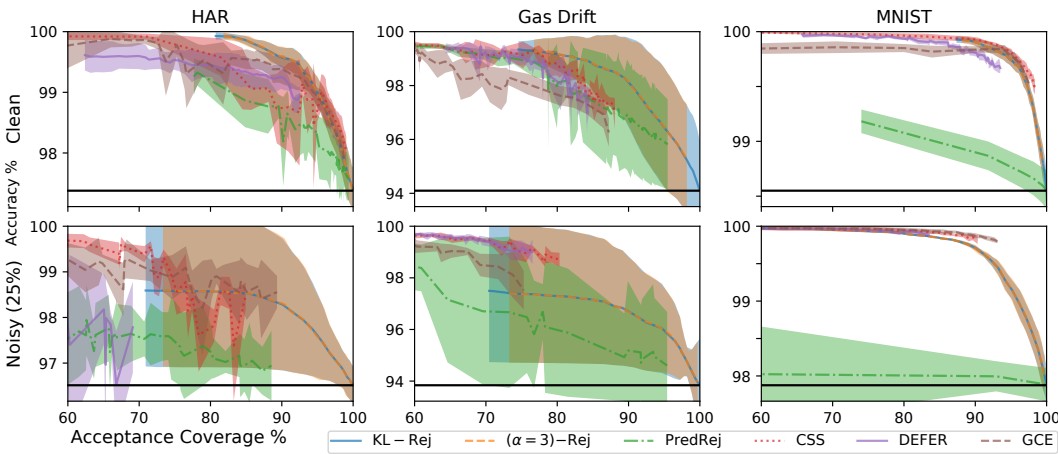

Figure 2: Accuracy vs coverage plots across select datasets and all approaches, with 50 equidistant $\tau \in (0, 1]$ and $c \in [0, 0.5)$ values (sorted by coverage). The black horizontal line depicts base models trained without rejection. Missing approaches in the plots indicates that the model rejects more than 60% of test points or has accuracy below the base model. Shaded region indicates $\pm 1$ s.t.d. region.

density ratio rejector are either competitive or superior to the baselines. One might notice that the aforementioned baselines do not or provide poor trade-offs for coverage values $> 95\%$ (as per Fig. 2). Indeed, to achieve rejection with high coverage (without architecture tuning), approaches which 'wrap' a base classifier seem preferable, *i.e.*, PredRej and our density ratios rejectors. Even at lower coverage targets (80%), Table 1 shows that density-ratio methods are comparable or superior in the more complex datasets of CIFAR-10 and the MedMNIST collection. If large models are allowed to be used for the rejector — as per the MNIST case — CSS, DEFER, and GCE can provide superior accuracy vs acceptance coverage trade-offs (noisy MNIST). However, this is not always true as per CIFAR-10 where all approaches are similarly effected by noise; or OctMNIST and OrgMNIST where approaches only slightly change with noise. The latter appears to be a consequence of label noise not effecting the Base classifier's accuracy (using the larger ResNet-18 architecture), as per Table 1.

Among the approaches which 'wrap' the base classifier $h$, we find that these approaches have higher variance ranges than the other approaches. In particular, the randomness of the base model potentially magnifies the randomness after rejection. The variance range of the base model tends to increase as the noise increases (additional ranges of noise for HAR and Gas Drift are presented in the Appendix). The influence on rejection is unsurprising as these 'wrapping' approaches predict via a composition of the original model (and hence inherits its randomness across folds). In general, our density ratio rejector outperforms PredRej. However, it should be noted that PredRej does not require a calibrated classifier. Among the density ratio rejectors, between KL and $\alpha = 3$, the only variation is in coverage region that the $\tau \in (0, 1]$ threshold covers. This follows from the fact that for similar distributions, $\varphi$-divergences act similarly [56, Theorem 7.20]. We find this pattern holds for other values of $\alpha$.

## 6   Limitations and Conclusions

We propose a new framework for rejection by learning idealized density ratios. Our proposed rejection framework links typically explored classification with rejection to generalized variational inference and distributionally robust optimization. It should be noted that although we have focused on classification, $L'(Q)$ could in theory be replaced by any other loss functions. In this sense, one could adapt this approach to other learning tasks such as regression, discussed in Appendix N. Furthermore, although we have focused on $\varphi$-divergences, there are many alternative ways idealized distribution can be constructed, *e.g.*, integral probability metrics [51, 9]. One limitation of our distributional way of rejecting is the reliance on approximating $P[Y \mid X]$ with model $h$. In future work, one may seek to approximate the density ratio $\rho$ by explicitly learning densities Q and P or via gradient based methods (for the latter, see Appendix O).

## Acknowledgments and Disclosure of Funding

We thank the reviewers and the area chair for the various suggestions which improved the paper. A majority of the work was done whilst AS was interning at Amazon.

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

# Supplementary Material

This is the Supplementary Material to Paper "Rejection via Learning Density Ratios". To differentiate with the numberings in the main file, the numbering of Theorems is letter-based (A, B, ...).

## Table of contents

**Proof**

**Deferred Content**

# A  Proof of Theorem 2.1

*Proof.* We first rewrite the coverage probability as an expectation:

$$\mathbb{E}_P[(1 - r(X)) \cdot \ell(Y, h(X))] + c \cdot \Pr[r(X) = 1] = \mathbb{E}_P[(1 - r(X)) \cdot \ell(Y, h(X)) + c \cdot r(X)].$$

We note that point-wise, the *proper* loss $\ell$ is minimized by taking the Bayes optimal classifier $\eta^\star(x) = \Pr[Y = +1 \mid X = x]$. Thus taking the argmin over all possible CPE classifiers, $h^\star = \eta^\star$. We note that the point-wise risk taken by the Bayes optimal classifier is typical denoted as the Bayes point-wise risk $\underline{L}(x) = \mathbb{E}_{P_{y|x=x}}[\ell(Y, \eta^\star(x))]$ [60, 61].

As such, we are left to minimize $r$ over,

$$\mathbb{E}_P[(1 - r(X)) \cdot \ell(Y, \eta^\star(X)) + c \cdot r(X)]$$
$$= \mathbb{E}_{P_x} \mathbb{E}_{P_{y|x=x}}[(1 - r(X)) \cdot \ell(Y, \eta^\star(X)) + c \cdot r(X)]$$
$$= \mathbb{E}_{P_x}[(1 - r(X)) \cdot \underline{L}(X) + c \cdot r(X)]$$
$$= \mathbb{E}_{P_x}[\underline{L}(X)] + \mathbb{E}_{P_x}[r(X) \cdot (c - \underline{L}(X))].$$

Thus, we immediately get the optimal $r^\star(x) = [\![\underline{L}(x) \geq c]\!]$. □

# B  Proof of Theorem 3.3

*Proof.* We use the theory of Lagrange multipliers to make different constraints explicit optimization problems.

Let us first consider the reduction from learing explicit distributions Eq. (8) to density ratios Eq. (9). First note that the objective Eq. (8) can be written as follows:

$$L(Q) + \lambda \cdot D_\varphi(P \parallel Q)$$
$$= \int L'(x) dQ(x) + \lambda \cdot \int \varphi\left(\frac{dQ}{dP}\right) dP(x)$$
$$= \int L'(x) \frac{dQ}{dP}(x) dP(x) + \lambda \cdot \int \varphi\left(\frac{dQ}{dP}(x)\right) dP(x)$$
$$= \mathbb{E}_P\left[L'(x) \cdot \frac{dQ}{dP}(x) + \lambda \cdot \varphi\left(\frac{dQ}{dP}(x)\right)\right]$$

The reduction to Eq. (9) now follows from a reduction from minimizing over $Q$ to $dQ/dP$, noting that $Q$ is restricted to be on the simplex. As such, the simplex constraints are transfered to:

$$\frac{dQ}{dP} \geq 0 \quad \text{and} \quad \int dP(x) \cdot \frac{dQ}{dP}(x) = \mathbb{E}_P\left[\frac{dQ}{dP}(X)\right] = 1,$$

where the former is the non-negativity of simplex elements and the latter is the normalization requirement. Hence, taking $\rho = \rho_\lambda \doteq \frac{dQ}{dP}$ completes the reduction. (we remove the subscript "$\lambda$" for the rest of the proof)

As such we have the optimization problem in Eq. (9), where we will convert the constraints into Lagrange multipliers, defining,

$$\mathcal{L}(\rho; a, b) = \mathbb{E}_P[\rho(X) \cdot L'(X) + \lambda \cdot \varphi(\rho(X))] - \int a'(x)\rho(x)dx + b \cdot (1 - \mathbb{E}_P[\rho(X)])$$
$$= \mathbb{E}_P[\rho(X) \cdot L'(X) + \lambda \cdot \varphi(\rho(X)) - a(X) \cdot \rho(X) - b \cdot \rho(X)] + b,$$

where $a(x) = a'(x) \cdot P(x)$.

We can obtain the first order optimality conditions by taking the functional derivative. Suppose that $\delta > 0$ and $h \colon \mathcal{X} \to \mathbb{R}$ is any function. The functional derivative is given by,

$$\frac{d}{d\delta}\mathcal{L}(\rho + \delta \cdot h; a, b)\Big|_{\delta=0}$$
$$= \mathbb{E}_P[h(X) \cdot (L'(X) + \lambda \cdot \varphi'(\rho(X) + \delta \cdot h(X)) - a(X) - b)]\Big|_{\delta=0}$$
$$= \mathbb{E}_P[h(X) \cdot (L'(X) + \lambda \cdot \varphi'(\rho(X)) - a(X) - b)].$$

Thus, the first order condition is,

$$0 = \frac{\mathrm{d}}{\mathrm{d}\delta}\mathcal{L}(\rho + \delta \cdot h; a, b)\bigg|_{\delta=0} = \mathbb{E}_{\mathrm{P}}\left[h(\mathsf{X}) \cdot (L'(\mathsf{X}) + \lambda \cdot \varphi'(\rho(\mathsf{X})) - a(\mathsf{X}) - b)\right],$$

for all $h: \mathfrak{X} \to \mathbb{R}$. As the condition must hold for all $h$ for an optimal $\rho^\star$, we have that for all $x \in \mathfrak{X}$,

$$0 = L'(x) + \lambda \cdot \varphi'(\rho^\star(x)) - a(x) - b$$

$$\iff \qquad \varphi'(\rho^\star(x)) = \frac{b + a(x) - L'(x)}{\lambda}$$

$$\iff \qquad \rho^\star(x) = (\varphi')^{-1}\left(\frac{b - L'(x) + a(x)}{\lambda}\right).$$

As required. $\qquad\qquad\qquad\qquad\qquad\qquad\qquad\qquad\qquad\qquad\qquad\qquad\qquad\qquad\qquad$ $\square$

## C   Proof of Corollary 3.4

*Proof.* First notice that the inner maximization in the DRO optimization Eq. (7) can be simplified as follows:

$$\sup_{\mathsf{Q} \in B_\varepsilon(\mathrm{P})} L(\mathsf{Q}) = -\left(\inf_{\mathsf{Q} \in B_\varepsilon(\mathrm{P})} -L(\mathsf{Q})\right)$$

$$= -\left(\inf_{\mathsf{Q} \in \mathcal{D}(\mathfrak{X})} \sup_{\lambda \geq 0} -L(\mathsf{Q}_\lambda) - \lambda \cdot (\varepsilon - D_\varphi(\mathrm{P} \parallel \mathsf{Q}_\lambda))\right)$$

$$= -\left(\sup_{\lambda \geq 0} \inf_{\mathsf{Q}_\lambda \in \mathcal{D}(\mathfrak{X})} -L(\mathsf{Q}_\lambda) - \lambda \cdot (\varepsilon - D_\varphi(\mathrm{P} \parallel \mathsf{Q}_\lambda))\right),$$

where the last inequality follows from Fan's minimax Theorem [25, Theorem 2] noting that $\mathsf{Q} \mapsto L(\mathsf{Q}_\lambda)$ is linear and the selected $\varphi$ per Definition 2.2 makes $\mathsf{Q} \mapsto D(\mathrm{P} \parallel \mathsf{Q}_\lambda)$ a convex lower semi-continuous function.

Now notice that the inner minimization of this simplification is exactly our idealized rejection distribution objective Eq. (8) when we negate the loss. As such, noticing that solutions to Eq. (8) are exactly given by $\mathrm{P} \cdot \rho$ yields the result after optimizing for the 'arguments' in the above DRO objective for both $\lambda$ and $\mathsf{Q}_\lambda$. $\qquad\qquad\qquad\qquad\qquad\qquad\qquad\qquad\qquad\qquad\qquad$ $\square$

## D   Proof of Corollary 4.1

We defer the proof of Corollary 4.1 to Appendix G, which covers any $\alpha$ including KL ($\alpha = 1$).

## E   Proof of Theorem 4.2

We breakdown Theorem 4.2 into two sub-theorems Theorem E.1 for each of the settings.

**Theorem E.1.** *Given the binary CPE setting presented in Theorem 2.1 and that we are given the optimal classifier $h^\star(x) = \mathrm{P}(\mathsf{Y} = 1 \mid \mathsf{X} = x)$, for any $\lambda > 0$ there exists a $\tau > 0$ such that the $r_\tau^{\mathrm{KL}}$ rejector generated the optimal density ratio in Corollary 4.1 is equivalent to the optimal rejector in Theorem 2.1.*

The proofs of each are similar. We first make the observation that the rejector function $r_\tau^{\mathrm{KL}}$ can be simplified as follows:

$$r_\tau^{\mathrm{KL}} = [\![L'(x) \leq -\lambda(\log Z + \log \tau)]\!],$$

Now we note that given a fixed $\lambda > 0$ (which also fixes $Z$), the RHS term has a one-to-one mapping from $\mathbb{R}_+$ to $\mathbb{R}$. Thus all that is to verify is that thresholding $L'(x)$ is equivalent to the rejectors of Theorems 2.1 and N.1.

*Proof.* For the CPE case, from assumptions, we have that $L'(x) = \mathbb{E}_{\mathsf{Y} \sim \mathrm{P}(\mathsf{Y}|\mathsf{X}=x)}[\ell(\mathsf{Y}, h^\star(x))] = \mathbb{E}_{\mathsf{Y} \sim h^\star(\mathsf{X}=x)}[\ell(\mathsf{Y}, h^\star(x))$ as $h^\star$ is optimal. Thus thresholding used in $r_\tau^{\mathrm{KL}}$ is equivalent to thresholding $L'(x)$ by keeping $\lambda$ fixed and changine $\tau$ appropriately. $\qquad\qquad\qquad\qquad$ $\square$

# F    Proof of Theorem 4.3

To prove the theorem, we will be using the standard Hoeffding's inequality [12].

**Theorem F.1** (Hoeffding's Inequality [12, Theorem 2.8]). *Let* $\mathsf{X}_1, \ldots, \mathsf{X}_n$ *be independent random variables such that* $\mathsf{X}_i$ *takes values in* $[a_i, b_i]$ *almost surely for all* $i \leq n$. *Defining* $\mathsf{X} = \sum_i \mathsf{X}_i - \mathbb{E}X_i$, *then for every* $t > 0$,

$$\Pr\left(\mathsf{X} \geq t\right) \leq \exp\left(-\frac{2t^2}{\sum_{i=1}^n (b_i - a_i)^2}\right).$$

*Proof.* Let us denote $Z$ to be the normalizer with the true expectation $\mathbb{E}_\mathrm{P}$ and $\hat{Z}$ to be the normalizer with the empirical expectation $\mathbb{E}_{\hat{\mathrm{P}}}$.

As $\ell$ is bounded, we take note of the following bounds,

$$\exp(-B/\lambda) \leq \max\left\{\exp\left(\frac{-L'(x)}{\lambda}\right), Z, \hat{Z}\right\} \leq \exp(B/\lambda).$$

This can be simply verified by taking the smallest and largest values of the $\exp$.

Now we simply have

$$
\begin{aligned}
|\rho^{\mathrm{KL}}(x) - \hat{\rho}^{\mathrm{KL}}(x)| &= \exp\left(\frac{-L'(x)}{\lambda}\right) \cdot \left|\frac{1}{Z} - \frac{1}{\hat{Z}}\right| \\
&\leq \exp\left(\frac{B}{\lambda}\right) \cdot \left|\frac{1}{Z} - \frac{1}{\hat{Z}}\right| \\
&= \exp\left(\frac{B}{\lambda}\right) \cdot \frac{|Z - \hat{Z}|}{|Z \cdot \hat{Z}|} \\
&\leq \exp\left(\frac{B}{\lambda}\right)^3 \cdot |Z - \hat{Z}|.
\end{aligned}
$$

Notice that $|Z - \hat{Z}|$ can be bounded via concentration inequality on (bounded) random variable $\exp\left(\frac{-L'(\mathsf{X})}{\lambda}\right)$.

Thus, we have that by Theorem F.1

$$\Pr\left(\left|\mathbb{E}_\mathrm{P}\left[\exp\left(\frac{-L'(x)}{\lambda}\right)\right] - \mathbb{E}_\mathrm{P}\left[\exp\left(\frac{-L'(x)}{\lambda}\right)\right]\right| > t\right) \leq 2\exp\left(\frac{-2 \cdot N \cdot t^2}{(b-a)^2}\right),$$

where $(b-a)^2 = (\exp(B/\lambda) - \exp(-B/\lambda))^2 = 4\sinh^2(B/\lambda)$.

Taking a union bound over $\mathcal{T}$, we have that

$$\Pr\left(\exists x \in \mathcal{T} : \left|\mathbb{E}_\mathrm{P}\left[\exp\left(\frac{-L'(x)}{\lambda}\right)\right] - \mathbb{E}_\mathrm{P}\left[\exp\left(\frac{-L'(x)}{\lambda}\right)\right]\right| > t\right) \leq 2M\exp\left(\frac{-N \cdot t^2}{2\sinh^2(B/\lambda)}\right).$$

Thus taking $t = \sinh(B/\lambda) \cdot \sqrt{\frac{2}{N} \cdot \log\left(\frac{2M}{\delta}\right)}$, we have that with probability $1 - \delta$ for $\delta > 0$, for any $x \in \mathcal{T}$,

$$
\begin{aligned}
|\rho^{\mathrm{KL}}(x) - \hat{\rho}^{\mathrm{KL}}(x)| &\leq \exp\left(\frac{B}{\lambda}\right)^3 \cdot |Z - \hat{Z}| \\
&\leq \exp\left(\frac{B}{\lambda}\right)^3 \cdot \sinh(B/\lambda) \cdot \sqrt{\frac{2}{N} \cdot \log\left(\frac{2M}{\delta}\right)}.
\end{aligned}
$$

As required.

$\square$

# G    Proof of Theorem 4.5

Before proving the theorems, we first state some basic properties of $\psi_\alpha$.

**Lemma G.1.** *For $\alpha \neq -1$, $\psi_\alpha(u \cdot v) = \psi_\alpha(u) \cdot \psi_\alpha(v)$ and $\psi_\alpha(1/v) = 1/\psi_\alpha(v)$. Furthermore, these statements hold when $\psi_\alpha$ is replaced with $\psi_\alpha^{-1}$.*

**Lemma G.2.**
$$\psi_\alpha^{-1}(z) = \begin{cases} z^{\frac{2}{1-\alpha}} & \text{if } \alpha \neq 1 \\ \exp(z) & \text{otherwise} \end{cases}. \tag{18}$$

The above statements follows directly from definition and simple calculation. The next statement directly connects $\varphi'_\alpha$ to $\psi_\alpha$.

**Lemma G.3.** *For $\alpha \neq 1$,*
$$(\varphi'_\alpha)(z) = \frac{2}{\alpha - 1} \cdot \psi_\alpha(1/z). \tag{19}$$

*Proof.* We prove this via cases.

- $\alpha = -1$:

$$\varphi'_{-1}(z) = \frac{\mathrm{d}}{\mathrm{d}z}(-\log z) = -z^{-1} = -1 \cdot z^{-\frac{1-\alpha}{2}} = \frac{2}{\alpha - 1} \cdot \psi_\alpha(1/z).$$

- $\alpha \neq \pm 1$:

$$\varphi'_\alpha(z) = \frac{\mathrm{d}}{\mathrm{d}z}\left(\frac{4}{1-\alpha^2} \cdot \left(1 - z^{\frac{1+\alpha}{2}}\right)\right) = -\frac{2}{1-\alpha}z^{\frac{\alpha-1}{2}} = -\frac{2}{1-\alpha} \cdot z^{-\frac{1-\alpha}{2}} = \frac{2}{\alpha - 1} \cdot \psi_\alpha(1/z)$$

$\square$

We now define the following constants depending on $\alpha$:

$$c_\alpha = \begin{cases} \psi_\alpha^{-1}\left(-\frac{2}{1-\alpha}\right) & \text{if } \alpha \neq 1 \\ \psi_\alpha^{-1}(-1) = \exp(-1) & \text{otherwise}. \end{cases} \tag{20}$$

**Lemma G.4.** *For $\alpha \neq 1$,*
$$(\varphi'_\alpha)^{-1}(z) = c_\alpha \cdot \frac{1}{\psi_\alpha^{-1}(z)}. \tag{21}$$

*Proof.* We prove this via cases.

- $\alpha = -1$:

$$\varphi'_{-1}(z) = \frac{\mathrm{d}}{\mathrm{d}z}(-\log z) = -z^{-1} = -1 \cdot z^{-\frac{1-\alpha}{2}} = -1 \cdot \psi_\alpha(1/z).$$

Then,

$$(\varphi'_{-1})^{-1} = \frac{1}{\psi_\alpha^{-1}(-1 \cdot z)} = \psi_\alpha^{-1}\left(\frac{1}{-1 \cdot z}\right) = \psi_\alpha^{-1}(-1) \cdot \frac{1}{\psi_\alpha^{-1}(z)} = c_\alpha \cdot \frac{1}{\psi_\alpha^{-1}(z)}.$$

- $\alpha \neq \pm 1$:

$$\varphi'_\alpha(z) = \frac{\mathrm{d}}{\mathrm{d}z}\left(\frac{4}{1-\alpha^2} \cdot \left(1 - z^{\frac{1+\alpha}{2}}\right)\right) = -\frac{2}{1-\alpha}z^{\frac{\alpha-1}{2}} = -\frac{2}{1-\alpha} \cdot z^{-\frac{1-\alpha}{2}} = -\frac{2}{1-\alpha} \cdot \psi_\alpha(1/z)$$

Then,

$$(\varphi'_\alpha)^{-1} = \frac{1}{\psi_\alpha^{-1}\left(-\frac{1-\alpha}{2} \cdot z\right)} = \frac{1}{\psi_\alpha^{-1}\left(-\frac{1-\alpha}{2}\right) \cdot \psi_\alpha^{-1}(z)} = \psi_\alpha^{-1}\left(-\frac{2}{1-\alpha}\right) \cdot \frac{1}{\psi_\alpha^{-1}(z)} = c_\alpha \cdot \frac{1}{\psi_\alpha^{-1}(z)}$$

$\square$

**Lemma G.5.** *For $\alpha = 1$,*

$$(\varphi'_\alpha)^{-1}(z) = c_\alpha \cdot \psi_\alpha^{-1}(z). \tag{22}$$

*Proof.* • $\underline{\alpha = 1}$:

$$\varphi'_1(z) = \frac{\mathrm{d}}{\mathrm{d}z}(z\log z) = \log z + 1$$

Then,

$$(\varphi'_{-1})^{-1} = \exp(z-1) = \exp(-1)\cdot\exp(z) = \psi_1^{-1}(-1)\cdot\psi_1^{-1}(z) = c_\alpha\cdot\psi_\alpha^{-1}(z)$$

$\square$

Thus now via Lemmas G.4 and G.5, we can prove the Theorems.

*Proof of Corollary 4.1 and Theorem 4.5.* The proof follows from utilizing either Lemmas G.4 and G.5 in conjunction with Theorem 3.3.

• $\underline{\alpha = 1}$: We have that,

$$\rho_\lambda^\varphi(x) = (\varphi')^{-1}\left(\frac{a(x) - L'(x) + b}{\lambda}\right)$$

$$= c_\alpha\cdot\psi_\alpha^{-1}\left(\frac{a(x) - L'(x) + b}{\lambda}\right)$$

$$= \exp\left(\frac{a(x) - L'(x) + b - 1}{\lambda}\right).$$

As $\rho_\lambda^\varphi(x)$ for $\alpha = 1$ is already positive by 'exp', by complementary slackness, the Lagrange multipliers $a(\cdot) = 0$. Hence, we can further simplify the above,

$$\rho_\lambda^\varphi(x) = \exp\left(\frac{-L'(x) + b - \lambda}{\lambda}\right).$$

The normalizer then can be easily calculated, renaming $b' = b - \lambda$, we simplify have that

$$\rho_\lambda^\varphi(x) = \exp\left(\frac{-L'(x) + b'}{\lambda}\right)$$

$$= \frac{1}{\exp(-b'/\lambda)}\cdot\exp\left(\frac{-L'(x)}{\lambda}\right),$$

which by normalization condition and setting $Z \doteq \exp(-b'/\lambda)$,

$$1 = \mathbb{E}\left[\frac{1}{Z}\cdot\exp\left(\frac{-L'(x)}{\lambda}\right)\right]$$

$$\Longleftrightarrow \quad Z = \mathbb{E}\left[\exp\left(\frac{-L'(x)}{\lambda}\right)\right],$$

which completes the case (Corollary 4.1).

• $\underline{\alpha \neq 1}$: We have that,

$$\rho_\lambda^\varphi(x) = (\varphi')^{-1}\left(\frac{a(x) - L'(x) + b}{\lambda}\right)$$

$$= c_\alpha\cdot\left(\psi_\alpha^{-1}\left(\frac{a(x) - L'(x) + b}{\lambda}\right)\right)^{-1}$$

$$\overset{(a)}{=} \psi_\alpha^{-1}\left(\frac{2}{\alpha - 1}\right)\cdot\left(\psi_\alpha^{-1}\left(\frac{a(x) - L'(x) + b}{\lambda}\right)\right)^{-1}$$

$$= \psi_\alpha^{-1}\left(\frac{2}{\alpha - 1}\right)\cdot\psi_\alpha^{-1}\left(\left(\frac{a(x) - L'(x) + b}{\lambda}\right)^{-1}\right)$$

$$= \psi_\alpha^{-1}\left(\frac{2}{\alpha - 1}\cdot\left(\frac{a(x) - L'(x) + b}{\lambda}\right)^{-1}\right),$$

where $(a)$ we exploit the that $(\psi_\alpha^{-1}(z))^{-1} = \psi_\alpha^{-1}(1/z)$ via Lemma G.1. $\square$

# H    Proof of Corollary 4.6

*Proof.* First we simplify the density ratio rejector.

$$\rho_\lambda^\alpha(x) = \psi_\alpha^{-1}\left(\frac{2}{\alpha-1}\cdot\left(a(x)-\frac{L'(x)}{\lambda}+b\right)^{-1}\right)$$

$$= \left(\frac{\alpha-1}{2}\cdot\left(a(x)-\frac{L'(x)}{\lambda}+b\right)\right)^{\frac{\alpha-1}{2}}.$$

We suppose that it is possible for

$$\left(\frac{\alpha-1}{2}\cdot\left(a(x)-\frac{L'(x)}{\lambda}+b\right)\right)^{\frac{\alpha-1}{2}} < 0$$

for values of $a(x)$, $b$, and $\lambda$. Otherwise, $\alpha(x) = 0$ and we are done due to the above equation's non-negativity.

Let $x \in \mathcal{X}$ be arbitrary. Suppose that $a(x) = 0$. Then by complementary slackness, we have that

$$\left(\frac{\alpha-1}{2}\cdot\left(-\frac{L'(x)}{\lambda}+b\right)\right)^{\frac{\alpha-1}{2}} > 0 \iff b - \frac{L'(x)}{\lambda} > 0.$$

By contra-positive, we have that $b - L'(x)/\lambda \le 0$ implies that $a(x) \ne 0$. By prime feasibility, in this case we also have $\rho(x) = 0$. We can solve either case by using the maximum as stated. $\square$

# I    Proof of Corollary 4.7

*Proof of Corollary 4.6.* The proof directly follows from a property of the $\alpha$-divergence when one of the measure have disjoint support. From [3] we have.

**Theorem I.1** ([3, Section 3.4.1 (4)]). *For $\alpha$-divergences, we have that*

*1. For $\alpha \le -1$, $D_\alpha(\mathrm{P}\parallel\mathrm{Q}) = \infty$ when $\mathrm{P}(x) \ne 0$ and $\mathrm{Q}(x) = 0$ for some $x \in \mathcal{X}$.*

This result immediately gives the result, as otherwise the objective function is $\infty$. $\square$

# J    Proof of Corollary 4.8

*Proof.* We first note simplifying $\rho_\lambda^{\alpha=3}$ via Corollary 4.6 yields:

$$\rho_\lambda^{\alpha=3}(x) = \max\left\{0, b - \frac{L'(x)}{\lambda}\right\}.$$

Thus our goal is to solve $b$ to give $\mathbb{E}_\mathrm{P}[\rho_\lambda^{\alpha=3}(\mathsf{X})] = 1$.

Now, consider the following,

$$1 + \frac{\mathbb{E}_\mathrm{P}[L'(\mathsf{X})]}{\lambda} - \frac{L'(x)}{\lambda} > 0 \iff \lambda > L'(x) - \mathbb{E}_\mathrm{P}[L'(\mathsf{X})],$$

where the latter holds uniformly for all $x$ from assumptions on $\lambda$.

Thus setting $b = 1 + \frac{\mathbb{E}_\mathrm{P}[L'(\mathsf{X})]}{\lambda}$, we simplify

$$\int \max\left\{0, 1 + \frac{\mathbb{E}_\mathrm{P}[L'(\mathsf{X})]}{\lambda} - \frac{L'(\mathsf{X})}{\lambda}\right\}\mathrm{dP}(x)$$

$$= \int 1 + \frac{\mathbb{E}_\mathrm{P}[L'(\mathsf{X})]}{\lambda} - \frac{L'(\mathsf{X})}{\lambda}\mathrm{dP}(x)$$

$$= 1.$$

As such, $b = 1 + \frac{\mathbb{E}_\mathrm{P}[L'(\mathsf{X})]}{\lambda}$ solves the required normalization. Substituting $b$ back into $\rho_\lambda^{\alpha=3}(x)$ yields the Theorem. $\square$

# K  Proof of Theorem 4.9

*Proof.* First we note that for any meas R, $\lambda > 2B$ implies that $\lambda > L'(x) - \mathbb{E}_{\mathrm{R}}[L'(\mathsf{X})]$ (taking largest and smallest values of $\ell$.

As such, for both $\mathrm{P}, \hat{\mathrm{P}}$ have closed forms Corollary 4.8

Thus, the bound can be simply shown to have,

$$|\rho_\lambda^{\alpha=3}(x) - \hat{\rho}_\lambda^{\alpha=3}(x)| = \frac{1}{\lambda} \cdot \left| \mathbb{E}_{\mathrm{P}}[L'(\mathsf{X})] - \mathbb{E}_{\hat{\mathrm{P}}}[L'(\mathsf{X})] \right|.$$

Thus by Hoeffding's inequality Theorem F.1 and union bound (see for instance the proof of Theorem 4.3), setting $t = B \cdot \sqrt{\frac{2}{N} \log\left(\frac{2M}{\delta}\right)}$, we have that with probability $1 - \delta$ for all $x \in \mathcal{T}$,

$$|\rho_\lambda^{\alpha=3}(x) - \hat{\rho}_\lambda^{\alpha=3}(x)| \leq \frac{B}{\lambda} \cdot \sqrt{\frac{2}{N} \log\left(\frac{2M}{\delta}\right)}.$$

As required. □

# L  Broader Impact

The paper presents work which reinterprets the classification with rejection problem in terms of learning distributions and density ratios. Beyond advancing Machine Learning in general, potential societal consequences include enhancing the understanding of the rejection paradigm and, consequentially, the human-in-the-loop paradigm. The general rejection setting aims to prevent models from making prediction when they are not confident, which can have societal significance when deployed in high stakes real life scenarios — allowing for human intervention.

# M  Distribution Generalization Bounds

In the following, we seek to provide generalization bounds on $\hat{\mathrm{Q}}_{\lambda,N} \doteq \hat{\mathrm{P}}_N \cdot \hat{\rho}_{\lambda,N}$. That is, one seeks to know that as $N \to \infty$ how and if $\hat{\mathrm{Q}}_{\lambda,N} \to \mathrm{Q}$. A natural measure of distance for probability measures is *total variation* [20],

$$\mathrm{TV}(\mathrm{P}, \mathrm{Q}) \doteq \frac{1}{2} \cdot \|\mathrm{P} - \mathrm{Q}\|_1 = \int |\mathrm{Q}(x) - \mathrm{P}(x)| \, \mathrm{d}x.$$

Let us also define

$$\|\mathrm{P} - \mathrm{Q}\|_\infty = \sup_{x \in \mathcal{X}} |\mathrm{Q}(x) - \mathrm{P}(x)|.$$

One immediately gets a rate if we assume that $|\mathcal{X}|$ is finite and we have a bound for $\hat{\rho}_{\lambda,N}$.

**Theorem M.1.** *Suppose that $|\mathcal{X}| = M < \infty$ is finite and bounded density ratio $|\hat{\rho}_{\lambda,N}| \leq B'$ for $B' > 0$. Then, if we have that with probability $1 - \delta$ that ,*

$$\|\rho(x) - \hat{\rho}_{N,\lambda}(x)\|_\infty \leq C \cdot \left( \sqrt{\frac{1}{N} \cdot \log \frac{2M}{\delta}} + C' \right),$$

*for some $C, C' > 0$, then we have that*

$$\mathrm{TV}(\mathrm{Q}, \hat{\mathrm{Q}}_{\lambda,N}) = \mathcal{O}\left( \sqrt{\frac{M^2 \log M}{N}} \right). \tag{23}$$

*Proof.* Noting that the empirical distribution is a sum of Dirac deltas $\hat{\mathrm{P}}_N(x) = \sum_{i \in [N]} \delta(x - x_i)$, we can establish a simple bound via a consequence of Hoeffding's Theorem Theorem F.1 (also noting that $\mathrm{P} = \mathbb{E}\hat{\mathrm{P}}_N$). We have that,

$$\Pr(|\mathrm{P}(x) - \hat{\mathrm{P}}_N(x)| \geq t) \leq 2 \exp\left(-2Nt^2\right).$$

Thus,

$$\Pr(\forall x \in \mathcal{X} : |\mathrm{P}_N(x) - \hat{\mathrm{P}}_N(x)| \geq t) \leq 2M \exp\left(-2Nt^2\right).$$

Setting $t = \sqrt{\frac{1}{2N} \log \frac{2M}{\delta}}$, we have with probability $1 - \delta$

$$\|\mathrm{P} - \hat{\mathrm{P}}_N\|_\infty \leq \sqrt{\frac{1}{2N} \log \frac{2M}{\delta}}.$$

We now consider the following:

$$\begin{aligned}
\|\mathrm{Q} - \hat{\mathrm{Q}}_{\lambda,N}\|_\infty &= \|\mathrm{P} \cdot \rho - \hat{\mathrm{P}}_N \cdot \hat{\rho}_{\lambda,N}\|_\infty \\
&= \|\mathrm{P} \cdot (\rho - \hat{\rho_{\lambda,N}}) + (\mathrm{P} - \hat{\mathrm{P}}_N) \cdot \hat{\rho}_{\lambda,N}\|_\infty \\
&\leq \|\mathrm{P} \cdot (\rho - \hat{\rho}_{\lambda,N})\|_\infty + \|(\mathrm{P} - \hat{\mathrm{P}}_N) \cdot \hat{\rho}_{\lambda,N}\|_\infty \\
&\leq \|(\rho - \hat{\rho}_{\lambda,N})\|_\infty + B' \cdot \|(\mathrm{P} - \hat{\mathrm{P}}_N)\|_\infty.
\end{aligned}$$

Taking a union bound of the above inequality and our assumption, we have that, for some $C > 0$, with probability $1 - \delta$,

$$\begin{aligned}
\|\mathrm{Q} - \hat{\mathrm{Q}}_{\lambda,N}\|_\infty &\leq \|(\rho - \hat{\rho}_{\lambda,N})\|_\infty + B' \cdot \|(\mathrm{P} - \hat{\mathrm{P}}_N)\|_\infty \\
&\leq C \cdot \sqrt{\frac{1}{N} \log \frac{4M}{\delta} + C'} + B' \cdot \sqrt{\frac{1}{2N} \log \frac{4M}{\delta}} \\
&= \mathcal{O}\left(\sqrt{\frac{\log M}{N}}\right).
\end{aligned}$$

Converting the bound to TV amounts to simply summing over $\mathcal{X}$, which gives $\mathrm{TV}(\mathrm{Q}, \hat{\mathrm{Q}}_{\lambda,N}) = \mathcal{O}(\sqrt{(M^2 \log M)/N})$, as required. $\qquad\square$

Notably, with appropriate assumptions, Theorems 4.3 and 4.9 can be used with Theorem M.1 to get bounds for $\hat{\mathrm{Q}}_{\lambda,N}^{\mathrm{KL}}$ and $\hat{\mathrm{Q}}_{\lambda,N}^{\alpha=3}$.

# N   Rejection for Regression

In the main-text of the paper we have focused on classification. However, many of the idea discussed can be extended for the regression setting. For instance, similar to Chow's Rule in Theorem 2.1 we can express the regression equivalent to the optimal solution of Eq. (2).

**Theorem N.1** (Optimal Regression Rejection [72]). *Let us consider the regression setting, where $\mathcal{Y} = \mathcal{Y}' = \mathbb{R}$ and $\ell(y, y') = \frac{1}{2}(y - y')^2$. Then w.r.t. Eq. (2), the optimal model is given by $h^\star(x) = \mathbb{E}_\mathrm{P}[\mathsf{Y} \mid \mathsf{X} = x]$ and the optimal rejector is given by $r^\star(x) = [\![\sigma^2(x) \leq c]\!]$, where $\sigma^2(x)$ is the conditional variance of $\mathsf{Y}$ given $\mathsf{X} = x$.*

For regression, there is no clear analogous notion of output "confidence score" unless the model explicitly outputs probabilities. Indeed, rejection method for regression explicitly requires the estimation of the target variable's conditional variance [72].

Similar to the CPE case, our KL density ratio rejector can provide a rejection policy equivalent to the typical case.

**Theorem N.2.** *Given the regression setting presented in Theorem N.1 and that we are given optimal regressor $h^\star(x) = \mathbb{E}_\mathrm{P}[\mathsf{Y} \mid \mathsf{X} = x]$, for any $\lambda > 0$ there exists a $\tau > 0$ such that the $r_\tau^{\mathrm{KL}}$ rejector generated the optimal density ratio in Corollary 4.1 is equivalent to the optimal rejector in Theorem N.1.*

*Proof.* The proof follows similarly to the proof of Theorem E.1. The regression case follows almost identically, by noticing that $L'(x) = \sigma^2(x)$ is the variance. This is similar to the proof of Theorem N.1 [72]. $\qquad\square$

Despite the equivalence, there is a difficult in using the density ratio rejectors, as per the closed form equations of Section 4, for regression. Estimating $L'(x) = \sigma^2(x)$ is challenging. Unlike classification where learning calibrated classifiers has a variety of approaches, learning a regression model which explicitly outputs probabilities is quite difficult. As such, approximating $P(Y \mid X = x)$ with the model $h(x)$ cannot be done.

## O   Gradient of Density Ratio Objective

As an alternative to the closed-form rejectors explored in the main-text, one may want to explore a method to learn $\rho$ iteratively. We consider the gradients of the optimization problem in Eq. (9). In practice, we found that we were unable to learn such rejectors via taking gradient updates and thus leave a concrete implementation of the idea for future work.

The idea comes from utilizing the "variational" aspect of the GVI formulation (which was not explored in the main-text). We suppose that the rejectors we are interested in come from a parameterized family. In particular, we consider the self-normalizing family $\rho_\vartheta / Z_\vartheta$, where $Z_\vartheta = \mathbb{E}_P[\pi_\vartheta(X)]$ normalizes the rejector such that $\mathbb{E}_P[\rho_\vartheta(X)] = 1$. Having the $Z_\vartheta$ normalizing term means that the constraint in Eq. (9) is satisfied for any $\vartheta$. The only constraint that we must have for $\pi_\vartheta$ is non-negativity, *i.e.*, $\pi_\vartheta$ is a neural network with exponential last activation functions from $\mathcal{X} \to \mathbb{R}_+$. By setting a parametric form of $\rho_\vartheta$, we implicitly restrict the set of idealized distributions to $Q_\vartheta = P \cdot \rho_\vartheta$. The gradients of such a parametric form can be calculated as follows.

**Corollary O.1.** *Let $\rho_\vartheta = \pi_\vartheta(x)/Z_\vartheta$. Then the gradient of Eq. (9) w.r.t. $\vartheta$ is given by,*

$$\mathbb{E}_{P_{x,y}} \left[ \nabla_\vartheta \left( \frac{\pi_\vartheta(X)}{Z_\vartheta} \right) \cdot \left( \varphi' \left( \frac{\pi_\vartheta(X)}{Z_\vartheta} \right) + \lambda \cdot \ell(Y, h(X)) \right) \right]. \tag{24}$$

*Proof.* The proof is immediate from differentiation of Eq. (9). $\qquad\square$

An alternative form of Eq. (24) can be found by noticing that $\nabla f = f \cdot \nabla \log f$. This provides an expression in terms of the log density ratio.

$$\mathbb{E}_{P_{x,y}} \left[ \frac{\pi_\vartheta(X)}{Z_\vartheta} \cdot \nabla_\vartheta \log \left( \frac{\pi_\vartheta(X)}{Z_\vartheta} \right) \cdot \left( \varphi' \left( \frac{\pi_\vartheta(X)}{Z_\vartheta} \right) + \lambda \cdot \ell(Y, h(X)) \right) \right].$$

One will notice that the gradient is in-fact the log-likelihood of the idealized distribution: noting P free of $\vartheta$, we have $\nabla_\vartheta \log(\pi_\vartheta(X)/Z_\vartheta) = \nabla_\vartheta \log(P(X) \cdot \pi_\vartheta(X)/Z_\vartheta) = \nabla_\vartheta \log Q_\vartheta$. As such, the gradient Eq. (24) is equivalent to the gradient of a weighted log-likelihood.

Despite the potential nice form of the gradient, we found that learning rejector through this was not possible. One limiting factor of computing such gradients is that we need to estimate $Z_\vartheta$ at each gradient calculation, *i.e.*, this must happen whenever $\vartheta$ changes. This can be particularly costly when $\mathcal{X}$ is high-dimensional. Secondly, we suspect that the model capacity of $\pi_\vartheta$ was not sufficient: we only tested on simple neural networks and convolutions neural networks, mirroring the architecture of the base classifiers used in our experimental setting.

## P   Finding the Best Rejection Cost

Both the baselines and density ratio approaches evaluated in Section 5 require a tuning of hyperparameters $c, \tau$. Practically, we are more interested in the rejection rate $P[r(X) = 1]$ rather than the abstract choices of $c, \tau$. Hence, one often wishes to select $c, \tau$ according to a coverage constraint.

One of the simplest choices of selecting $c, \tau$ is to utilize a calibration set and approximate the rejection rate / coverage on this data. Our experiments in Section 5 mirrors this in Tables 1 and I by treating the final test set as the calibration data.

Other works in the literature utilize more sophisticated methods for deriving coverage constraints. [29] picks threshold values which provide a guarantee on a coverage-modified risk requirement. Another approach [57] does not require an additional calibration set and instead fits threshold values $\tau$ by exploiting stacking confidences and (stratified) cross-fitting. Although one could consider such approaches for all baselines, including those which utilize the cost-model of rejection via $c$, one will

have to pay a price in retraining the learned rejector $r$ for each instance of the hyperparameter $c$ that is searched. As a result, approaches which can trade-off accuracy and coverage via a threshold variable $\tau$ (rather than an expensive retraining) are computationally preferable when exhaustively searching for a good $c, \tau$.

We leave more sophisticated methods for selecting $\tau$ for density ratio rejectors for future work.

## Q   Additional Experimental Details

### Q.I   Training Settings

The neural network architecture used to train the base classifiers and baselines are almost identical. For the baseline approaches which have output dimension which is different than the output of the original neural network, we modify the last linear layer of the base classifier's architecture to fit the baseline's requirements, *e.g.*, adding an additional output dimension for rejection in DEFER. Our architectures utilize batch normalization [38] and dropout [65] in a variety of places. Training settings are mostly identical, with some baselines requiring slight changes.

The base model's architecture is as follows.

- HAR (CC BY 4.0): We utilize a two hidden layer neural network with batch normalization. Both hidden layer is $64$ neurons and the activation function is the sigmoid function. We take a 64 batch size, 40 training epochs and a 0.0001 learning rate.

- Gas Drift (CC BY 4.0): We utilize a two hidden layer neural network with batch normalization. Both hidden layers are $64$ neurons and the activation function is the sigmoid function. We take a 64 batch size, 40 training epochs and a 0.0001 learning rate.

- MNIST (CC BY-SA 3.0): We utilize a convolutional neural network with two convolutional layers and two linear layers. The architecture follows directly from the MNIST example for PyTorch. We utilize the sigmoid function activation function. We take a 256 batch size, 40 training epochs and a 0.0001 learning rate.

- CIFAR-10 (CC BY 4.0): We utilize a ResNet-18 classifier. Random cropping and horizontal flipping data augmentation is utilized in training. We take a 256 batch size, 40 training epochs and a 0.0001 learning rate.

- OrgMNIST (CC BY 4.0): We utilize a ResNet-18 classifier. We take a 256 batch size, 40 training epochs and a 0.0001 learning rate.

- OctMNIST (CC BY 4.0): We utilize a ResNet-18 classifier. We take a 256 batch size, 40 training epochs and a 0.0001 learning rate.

For CSS, as noted in [53], batch normalization is needed at the final layer to stabilize training.

All training utilizes the Adam [41] optimizer.

All datasets we consider are in the public domain, *e.g.*, UCI [6].

### Q.II   Extended Table and Plots

Table I presents an extended version of Table 1 over coverage targets of 80% and 90%. The standard deviation is additionally reported. One can see the observation from the main text are consistent with this extended table.

Plots Figs. I to III show Fig. 2 over and extend region of acceptable coverage percentages. In addition, we include a larger range of noise rates for HAR and Gas Drift. For MNIST, we explore a larger range of noise rates for MNIST in Fig. VII for our density ratio rejectors. We find that the findings in the main text are extended to these additional noise rates. In particular, we find that the our density ratio rejectors can be competitive with the various baseline approaches. We find that our density ratio approaches can have more fine-grained trade-offs at higher ranges of acceptance coverage. This is an important region where the budge for rejection may be low (and only a few examples, *e.g.* $< 10\%$, can be rejected). Indeed, the baseline approaches which do not 'wrap' a base model require a lower *maximum acceptance coverage* as the noise rate increases (the approaches require a higher rejection % for any type of rejection). Nevertheless, we do see a downside of the density ratio approach: the

Table I: Extended Summary performance of rejection methods over all baselines and datasets targeting 80% and 90% coverage. Each cell reports the "accuracy (accuracy s.t.d.)[coverage (coverage s.t.d.)]" values.

| | | | Base | KL-Rej | $(\alpha=3)$-Rej | PredRej | CSS | DEFER | GCE |
|---|---|---|---|---|---|---|---|---|---|
| Target Coverage: 80% | Clean | HAR | 97.38 (0.34) [100.00 (0.00)] | 99.93 (0.06) [80.74 (0.84)] | 99.93 (0.06) [81.82 (0.74)] | 98.86 (0.44) [84.86 (8.45)] | 99.58 (0.31) [81.39 (1.48)] | 99.44 (0.19) [80.16 (1.54)] | 99.31 (0.52) [81.78 (4.69)] |
| | | Gas Drift | 94.10 (1.87) [100.00 (0.00)] | 99.16 (0.62) [80.24 (0.75)] | 99.16 (0.62) [80.24 (0.75)] | 98.12 (1.03) [80.09 (6.03)] | 98.68 (0.39) [80.33 (1.18)] | 98.06 (0.97) [80.49 (1.09)] | 97.62 (0.36) [80.43 (3.92)] |
| | | MNIST | 98.55 (0.19) [100.00 (0.00)] | 99.93 (0.03) [87.37 (0.66)] | 99.93 (0.03) [88.08 (0.58)] | 99.18 (0.11) [74.03 (4.05)] | 99.95 (0.03) [83.21 (0.89)] | 99.93 (0.01) [80.38 (1.60)] | 99.85 (0.04) [80.02 (3.75)] |
| | | CIFAR-10 | 90.20 (0.29) [100.00 (0.00)] | 97.22 (0.18) [80.27 (0.23)] | 97.17 (0.17) [80.53 (0.21)] | 91.40 (0.89) [74.42 (16.73)] | 95.45 (0.95) [80.56 (1.14)] | 93.72 (0.94) [80.57 (4.78)] | 94.25 (0.31) [80.91 (0.83)] |
| | | OrganMNIST | 89.10 (1.06) [100.00 (0.00)] | 96.55 (0.78) [80.25 (1.42)] | 96.52 (0.82) [80.33 (1.18)] | 93.79 (0.95) [81.77 (6.86)] | 94.49 (0.65) [80.17 (4.32)] | 93.47 (0.55) [80.16 (4.59)] | 93.68 (1.73) [80.04 (2.48)] |
| | | OctMNIST | 91.93 (0.22) [100.00 (0.00)] | 97.08 (0.20) [80.94 (0.28)] | 97.18 (0.19) [80.22 (0.20)] | 93.43 (0.81) [85.57 (8.22)] | 95.40 (1.06) [81.05 (2.58)] | 94.66 (0.31) [85.32 (4.92)] | 94.91 (0.79) [86.67 (3.26)] |
| | Noisy (25%) | HAR | 96.51 (0.44) [100.00 (0.00)] | 98.56 (1.66) [81.16 (15.40)] | 98.56 (1.66) [81.13 (15.41)] | 97.22 (0.55) [80.24 (7.73)] | 97.82 (0.35) [80.50 (0.67)] | 97.78 (1.00) [69.11 (6.36)] | 98.85 (0.43) [80.18 (3.11)] |
| | | Gas Drift | 93.84 (1.81) [100.00 (0.00)] | 97.30 (2.60) [80.02 (16.41)] | 97.28 (2.59) [80.09 (16.38)] | 95.87 (2.48) [81.51 (9.20)] | 98.71 (0.27) [80.32 (1.91)] | 99.02 (0.48) [76.54 (2.11)] | 97.52 (0.75) [75.31 (2.76)] |
| | | MNIST | 97.88 (0.23) [100.00 (0.00)] | 99.89 (0.01) [80.44 (1.36)] | 99.89 (0.01) [80.64 (1.36)] | 98.00 (0.20) [92.95 (10.17)] | 99.94 (0.03) [80.38 (1.08)] | 99.93 (0.03) [81.45 (0.58)] | 99.95 (0.03) [81.20 (1.55)] |
| | | CIFAR-10 | 85.31 (0.18) [100.00 (0.00)] | 92.25 (0.33) [81.61 (0.98)] | 92.50 (0.38) [80.71 (1.00)] | 85.84 (1.17) [88.25 (23.07)] | 89.58 (1.28) [81.83 (2.48)] | 90.93 (0.45) [80.29 (1.59)] | 92.22 (0.12) [80.55 (0.41)] |
| | | OrganMNIST | 89.10 (0.77) [100.00 (0.00)] | 95.86 (1.36) [82.07 (1.95)] | 96.29 (0.94) [80.93 (1.16)] | 93.40 (0.74) [83.78 (5.13)] | 96.74 (0.68) [81.19 (0.77)] | 94.67 (1.17) [80.39 (5.45)] | 94.48 (0.77) [80.39 (3.51)] |
| | | OctMNIST | 91.89 (0.20) [100.00 (0.00)] | 97.17 (0.21) [80.29 (0.87)] | 97.10 (0.21) [80.72 (0.78)] | 93.42 (1.10) [83.14 (12.31)] | 95.49 (1.35) [81.20 (4.24)] | 94.08 (0.64) [89.10 (2.42)] | 94.63 (1.14) [77.56 (13.79)] |
| Target Coverage: 90% | Clean | HAR | 97.38 (0.34) [100.00 (0.00)] | 99.56 (0.17) [90.34 (0.38)] | 99.58 (0.17) [90.19 (0.42)] | 98.19 (0.74) [90.20 (14.01)] | 98.75 (0.73) [90.89 (1.26)] | 99.23 (0.19) [90.34 (0.29)] | 99.37 (0.24) [90.27 (0.75)] |
| | | Gas Drift | 94.10 (1.87) [100.00 (0.00)] | 98.05 (1.71) [90.24 (1.58)] | 98.12 (1.58) [90.06 (1.37)] | 96.64 (2.06) [90.30 (6.65)] | 96.96 (0.56) [88.09 (0.70)] | 97.46 (0.44) [84.51 (2.08)] | 96.26 (0.53) [87.24 (2.24)] |
| | | MNIST | 98.55 (0.19) [100.00 (0.00)] | 99.89 (0.04) [90.57 (0.55)] | 99.89 (0.04) [90.84 (0.53)] | 98.87 (0.13) [91.82 (1.52)] | 99.93 (0.02) [90.42 (0.36)] | 99.80 (0.03) [90.22 (0.39)] | 99.84 (0.03) [90.72 (3.41)] |
| | | CIFAR-10 | 90.20 (0.29) [100.00 (0.00)] | 94.41 (0.21) [90.19 (0.19)] | 94.43 (0.21) [90.14 (0.20)] | 90.32 (0.22) [93.66 (10.05)] | 92.77 (0.68) [90.16 (0.72)] | 92.98 (0.86) [91.01 (1.63)] | 91.54 (0.44) [90.43 (0.47)] |
| | | OrganMNIST | 89.10 (1.06) [100.00 (0.00)] | 92.60 (0.87) [90.15 (1.43)] | 92.57 (0.94) [90.27 (1.20)] | 91.44 (1.68) [90.49 (7.46)] | 92.52 (0.66) [90.42 (1.78)] | 92.35 (0.58) [90.16 (1.68)] | 90.88 (0.75) [90.07 (2.00)] |
| | | OctMNIST | 91.93 (0.22) [100.00 (0.00)] | 94.98 (0.26) [91.35 (0.43)] | 95.17 (0.25) [90.69 (0.38)] | 92.23 (0.60) [96.63 (4.21)] | 91.96 (1.94) [91.50 (1.62)] | 93.49 (0.46) [91.18 (2.08)] | 94.20 (0.68) [90.38 (2.12)] |
| | Noisy (25%) | HAR | 96.51 (0.44) [100.00 (0.00)] | 98.28 (1.44) [90.11 (8.10)] | 98.29 (1.45) [90.40 (7.86)] | 96.94 (0.77) [88.55 (4.82)] | 98.48 (0.29) [84.94 (1.07)] | 97.78 (1.00) [69.11 (6.36)] | 98.56 (0.35) [89.34 (1.23)] |
| | | Gas Drift | 93.84 (1.81) [100.00 (0.00)] | 96.58 (2.05) [90.75 (7.96)] | 96.68 (2.11) [90.04 (8.34)] | 95.11 (1.85) [91.71 (6.72)] | 98.71 (0.27) [80.32 (1.91)] | 99.02 (0.48) [76.54 (2.11)] | 97.52 (0.75) [75.31 (2.76)] |
| | | MNIST | 97.88 (0.23) [100.00 (0.00)] | 99.71 (0.03) [90.01 (0.99)] | 99.70 (0.04) [90.03 (0.89)] | 98.00 (0.20) [92.95 (10.17)] | 99.82 (0.06) [90.33 (0.61)] | 99.89 (0.03) [83.47 (0.43)] | 99.86 (0.02) [90.81 (0.85)] |
| | | CIFAR-10 | 85.31 (0.18) [100.00 (0.00)] | 89.43 (0.39) [90.44 (0.94)] | 89.42 (0.39) [90.43 (0.92)] | 85.66 (0.69) [91.06 (17.80)] | 89.71 (1.01) [83.40 (1.44)] | 89.94 (0.92) [83.03 (1.17)] | 92.14 (0.24) [81.90 (1.23)] |
| | | OrganMNIST | 89.10 (0.77) [100.00 (0.00)] | 92.56 (0.62) [90.22 (1.70)] | 92.53 (0.64) [90.29 (1.60)] | 92.24 (0.76) [90.61 (2.35)] | 91.92 (1.05) [90.22 (1.17)] | 91.77 (0.94) [90.11 (2.06)] | 92.17 (0.24) [90.91 (0.45)] |
| | | OctMNIST | 91.89 (0.20) [100.00 (0.00)] | 95.22 (0.21) [90.47 (0.52)] | 95.23 (0.23) [90.45 (0.45)] | 92.41 (0.36) [90.70 (8.09)] | 93.28 (0.52) [91.49 (0.93)] | 93.63 (0.40) [90.89 (3.05)] | 93.39 (0.89) [90.22 (0.89)] |

quality of the density ratio rejectors is dependent on the initial model. As such, at higher levels of noise there can be higher variation in the quality of rejection, see Fig. II. Interestingly, for MNIST and CIFAR-10, Figs. III and IV, the base model the density ratio rejectors are more robust across noise rates than other models. This seems to be due to the default MNIST architecture being robust against higher noise rates (notice that the s.t.d. range is also quite small at 100% coverage). In OctMNIST and OrganMNIST, Figs. V and VI, we find that there is little to no change in performance, likely due to the Base classifier's own performance only changing slightly with the addition of noise.

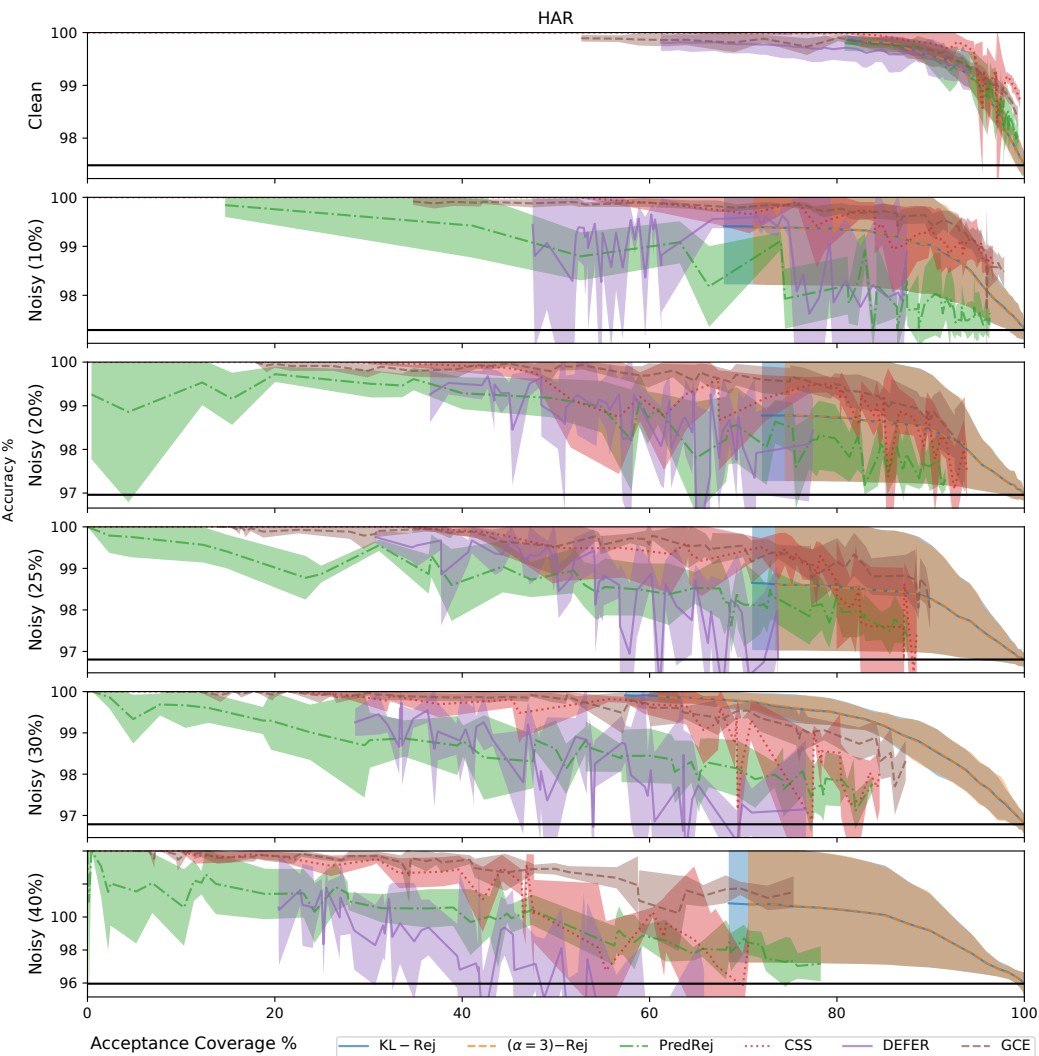

Figure I: Extended plots for HAR of Fig. 2.

## Q.III  Smaller models case study

In the following, we consider the Gas Drift dataset when models are switched to a base model with only a single hidden layer. First we make note of the original setting explored in the main text. In Table II, we take note of the number of tunable parameters in all approaches and baselines. Notably, these default parameter / architecture sizes are similar to [14], with the HAR and Gas Drift setting including an additional hidden layer than previously utilized in the literature.

In Table III, we note the setting we consider in this subsection. The parameter sizes of the Gas Drift dataset is reduced to the originally explored model sizes in [14]. Notice that both the base models and the baseline approaches have reduced parameter sizes. It should be noted that this smaller parameter

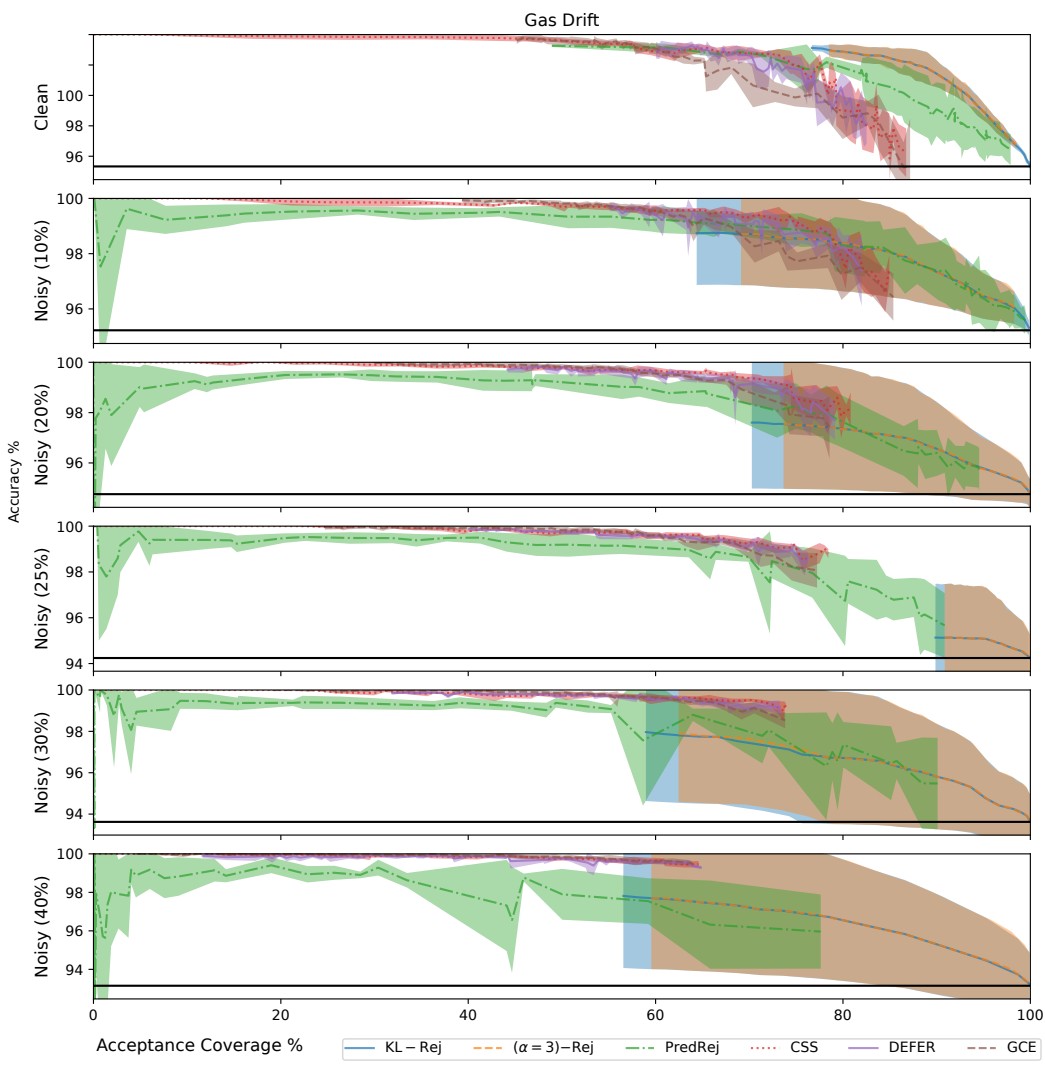

Figure II: Extended plots for Gas Drift of Fig. 2.

Table II: Default parameter sizes of experiments.

| Dataset | BaseClf | $\alpha-$Rej | DEFER | GCE | CSS | PredRej |
|---------|---------|---------|-------|-----|-----|---------|
| HAR | 40,647 | 40,648 | 40,646 | 40,711 | 40,711 | 80,968 |
| Gas Drift | 12,935 | 12,936 | 12,934 | 12,999 | 12,999 | 25,544 |
| MNIST | 1,199,883 | 1,199,884 | 1,200,138 | 1,200,011 | 1,200,267 | 2,398,860 |
| CIFAR-10 | 21,282,123 | 21,282,124 | 21,283,146 | 21,282,635 | 21,283,659 | 42,560,652 |
| OctMNIST | 11,169,733 | 11,169,734 | 11,170,756 | 11,170,245 | 11,171,269 | 22,338,950 |
| OrganMNIST | 11,173,324 | 11,173,325 | 11,174,347 | 11,173,836 | 11,174,860 | 22,342,541 |

size setting can be useful in the related learning with deferral setting [52], where having a small model to defer to a larger model is needed.

The results are reported in Figs. VIII and IX. We can see that in this setting, PredRej and our density ratio approaches are more competitive. This might indicate that for simple base models, approaches which 'wrap' a base model for rejection can be quite effective (especially in higher coverage regimes). In general, it seems with this smaller architecture regime, the 'non-wrapping' baseline approaches only provide rejection options when the acceptance coverage is lower than 70%.

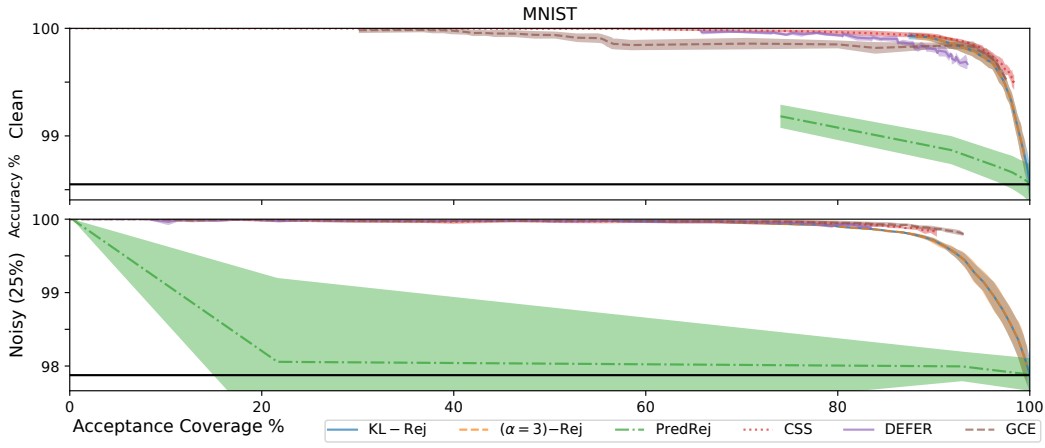

Figure III: Extended plots for MNIST of Fig. 2.

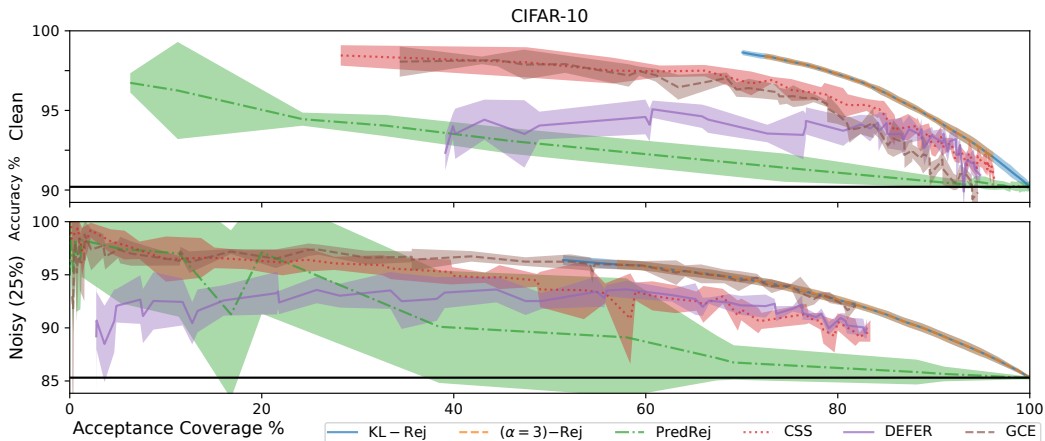

Figure IV: Deferred extended plots for CIFAR-10 of Fig. 2.

## Q.IV Parameter sweeps over $\alpha$ and $\lambda$

The following shows parameter sweeps over $\alpha$ and $\lambda$ for our density ratio rejectors. These are given by Figs. X and XI respectively. We find that increasing $\alpha$ compresses the trade-off curve from both sides. While decrease $\lambda$ extends the trade-off curve on the left side.

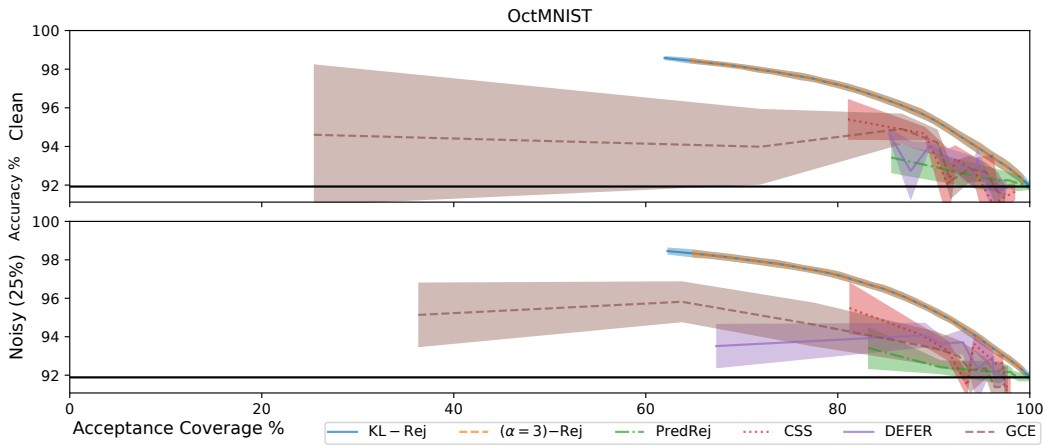

Figure V: Deferred extended plots for OctMNIST of Fig. 2.

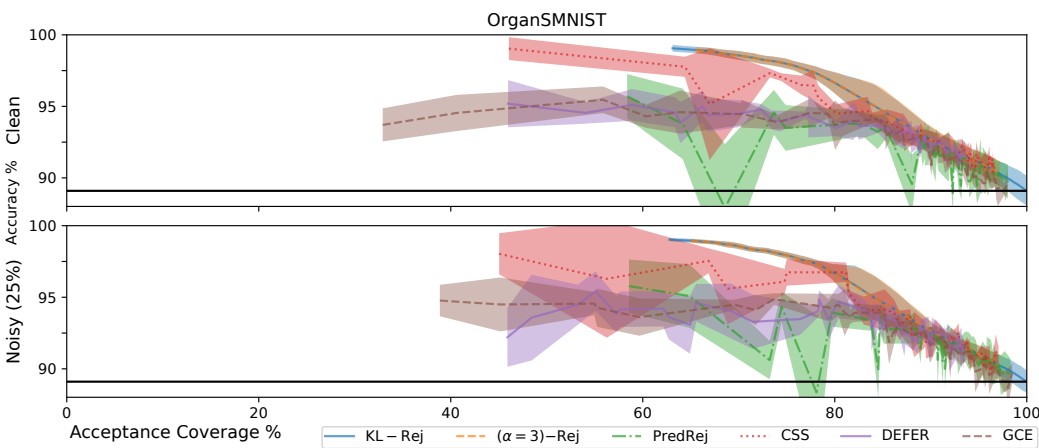

Figure VI: Deferred extended plots for OrganMNIST of Fig. 2.

Table III: Alternative parameter sizes of experiments.

| Dataset | BaseClf | $\alpha-$Rej | DEFER | GCE | CSS | PredRej |
|---|---|---|---|---|---|---|
| HAR | 36,487 | 36,488 | 36,486 | 36,551 | 36,551 | 72,648 |
| Gas Drift | 8,775 | 8,776 | 8,774 | 8,839 | 8,839 | 17,224 |

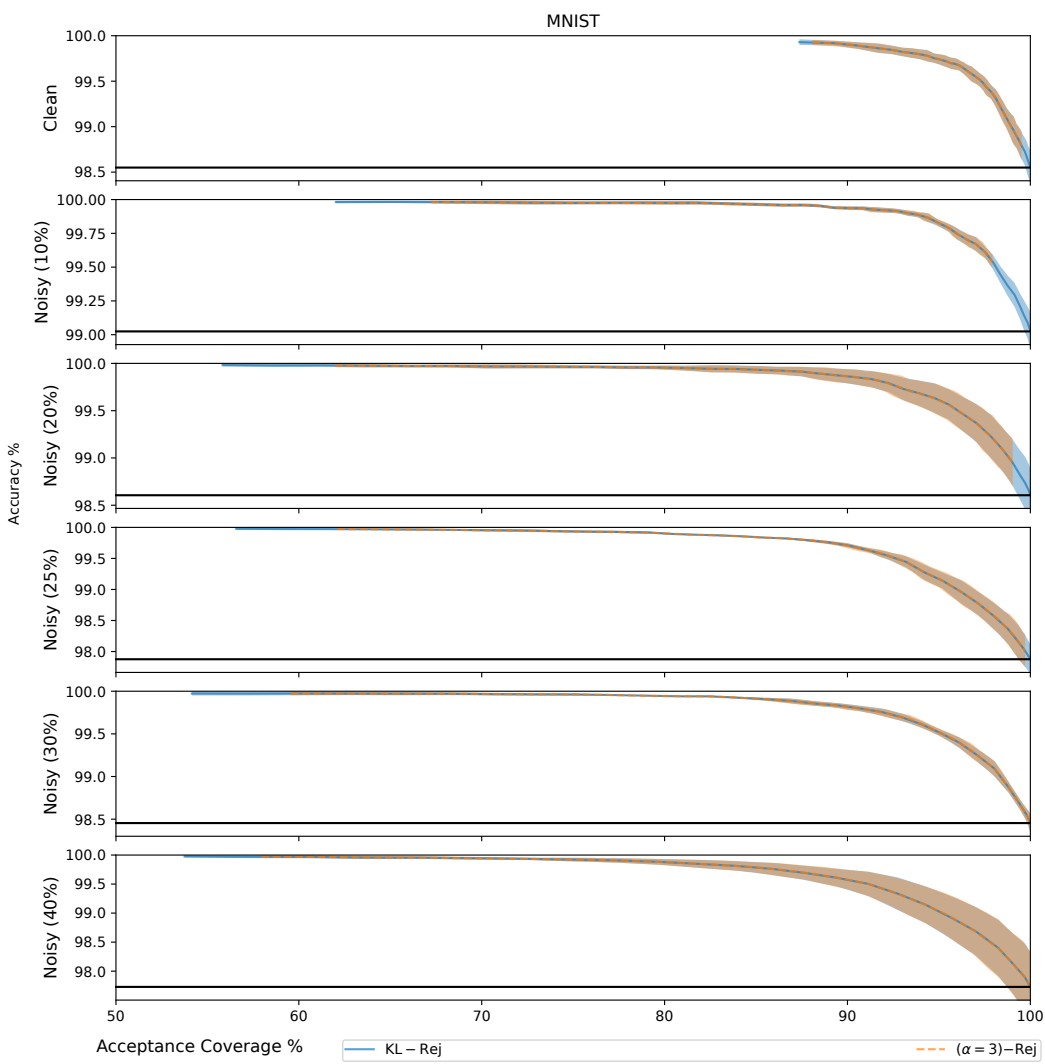

Figure VII: MNIST with different noises for density ratio approaches.

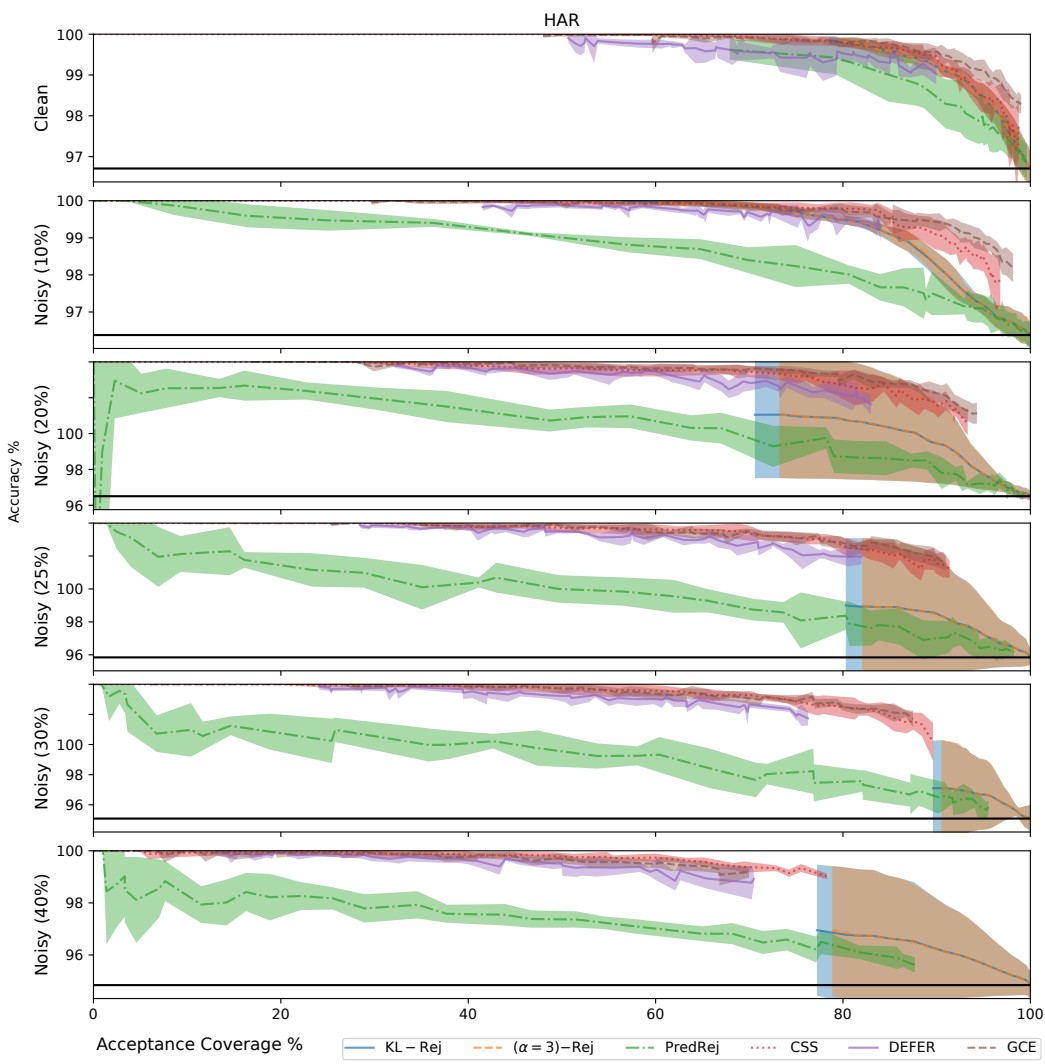

Figure VIII: Plots for HAR with smaller models.

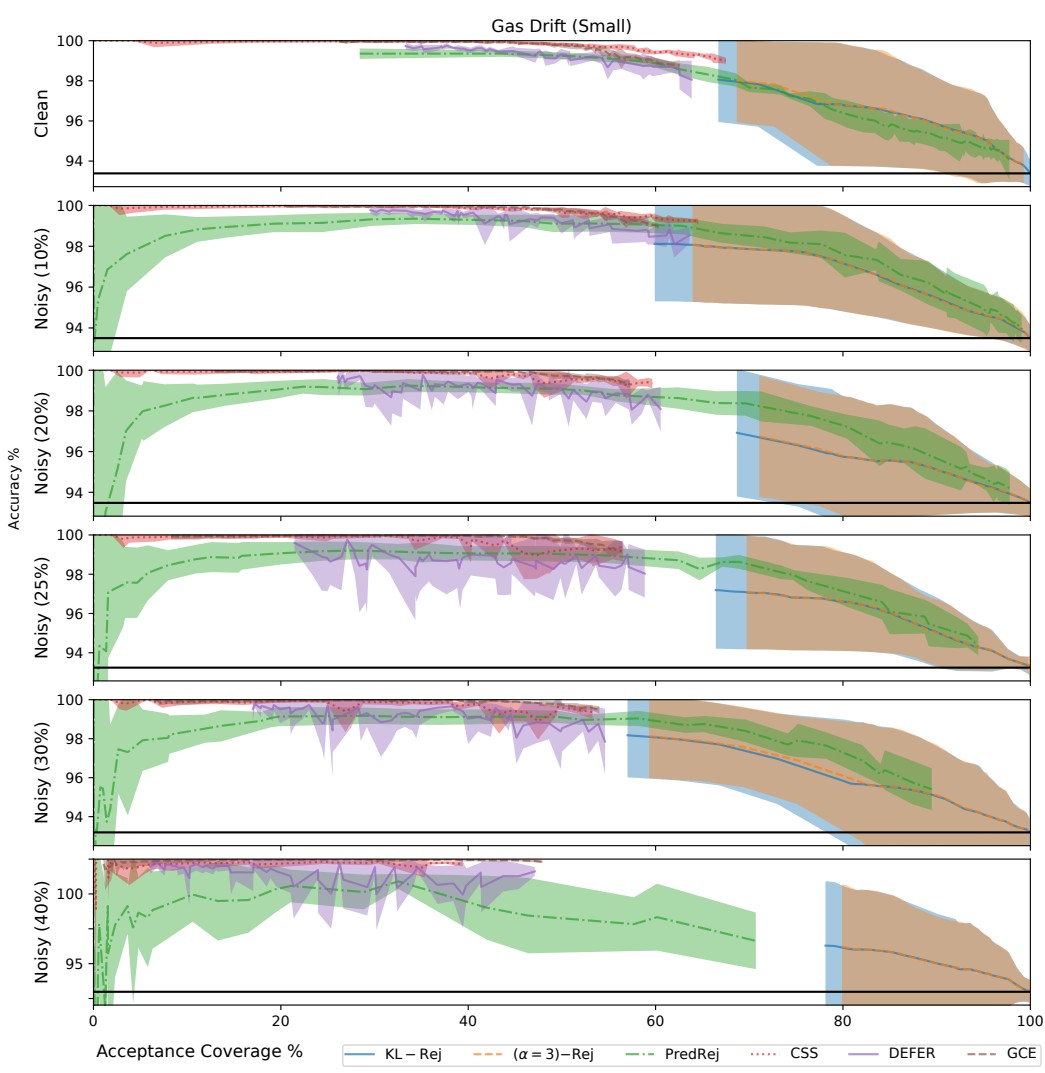

Figure IX: Plots for Gas Drift with smaller models.

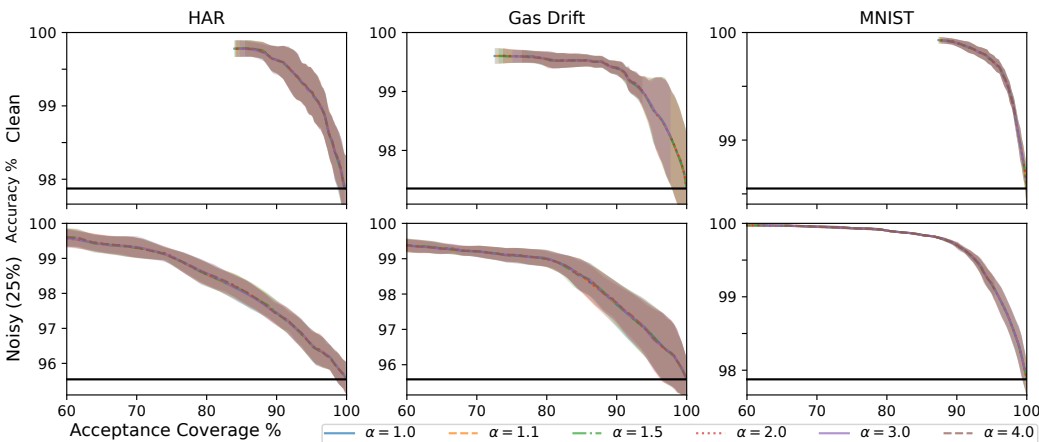

Figure X: $\alpha$ parameter sweep over dataset + noise combinations. S.t.d. shade is not utilized, but instead end points of the trade-off curves for $\tau \in (0, 1]$ are shown via vertical bars

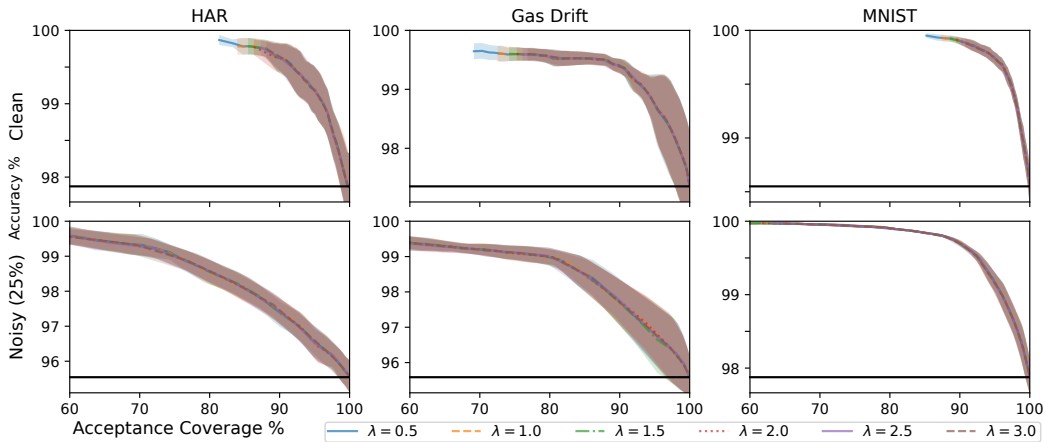

Figure XI: $\lambda$ parameter sweep over dataset + noise combinations for KL rejector. S.t.d. shade is not utilized, but instead end points of the trade-off curves for $\tau \in (0, 1]$ are shown via vertical bars

