# OpenReview forum: "Rejection via Learning Density Ratios"
_NeurIPS.cc/2024/Conference — NeurIPS 2024 poster_

### Official Review · Reviewer_dzAs · 2024-07-02

**Soundness:** 2
**Presentation:** 3
**Contribution:** 3
**Rating:** 6
**Confidence:** 3

**Summary:**

The paper proposes a distributional perspective to build abstaining classifiers. By considering an idealized distribution, the authors show that this translates into optimizing a loss risk with a $\varphi$-divergence regularization term. Moreover, they provide results when considering $\alpha$-divergences, a specific family of $\varphi$-divergences. Empirical evaluation is performed over clean and noisy datasets.

**Strengths:**

The main strengths of the paper are:

i) the paper is clearly written, with clear contributions advancing the state of the art in abstaining classifiers
ii) the theoretical contributions seem sound;
iii) the proposed method is original and bridges DROs with Learning to Reject.

**Weaknesses:**

The main weaknesses of the paper are:

i) there is no related work section explicitly dedicated to learning with a reject option. See, e.g., [a] for a recent survey.


ii) the empirical evaluation can be improved:
* for instance, the paper considers only four baselines (mainly from Learning to Defer literature) and ignores popular alternatives such as softmax response [26]
* the method is tested only on three datasets, which seem easy to solve. In particular, MNIST might be too easy to evaluate abstaining classifiers (an accuracy of 100% is easy to achieve with relatively few rejections). A more interesting dataset collection could be MedMNIST [b], which contains medical images from real data tasks.



[a] Hendrickx, Kilian, Lorenzo Perini, Dries Van der Plas, Wannes Meert, and Jesse Davis. "Machine learning with a reject option: A survey." Machine Learning 113, no. 5 (2024): 3073-3110.\
[b] Yang, Jiancheng, Rui Shi, and Bingbing Ni. "Medmnist classification decathlon: A lightweight automl benchmark for medical image analysis." In 2021 IEEE 18th International Symposium on Biomedical Imaging (ISBI), pp. 191-195. IEEE, 2021.

**Questions:**

I have a few questions for the authors:

1) is there a reason why you did not compare with the simplest (and often the best) abstaining classifier, i.e. the softmax response? In [c], the authors show that the classifier built using this strategy can be optimal. Moreover, recent empirical works [d,e] show that score-based approaches outperform other abstaining classifiers.

2) how did you train DEFER, without access to human predictions? As far as I know, the surrogate loss also requires specific human predictions; however, I could not find in the Appendix how this is done in practice.

[c] Franc, Vojtech, Daniel Prusa, and Vaclav Voracek. "Optimal strategies for reject option classifiers." Journal of Machine Learning Research 24, no. 11 (2023): 1-49.\
[d] Jaeger, Paul F., Carsten Tim Lüth, Lukas Klein, and Till J. Bungert. "A Call to Reflect on Evaluation Practices for Failure Detection in Image Classification." In The Eleventh International Conference on Learning Representations. (2023)\
[e] Pugnana, Andrea, Lorenzo Perini, Jesse Davis, and Salvatore Ruggieri. "Deep neural network benchmarks for selective classification." arXiv preprint arXiv:2401.12708 (2024).

**Limitations:**

The authors discuss the limitations of their approach.

---

> ### Author Rebuttal · Authors · 2024-08-07
>
> We thank the reviewer for sharing their insights and providing useful suggestions on improving our evaluation. Please find the rebuttal below.
>
> > Related Work
>
> We thank the reviewer for their various references. We will make sure to include these in the next version of the paper. We would also be happy to add a dedicated related work section. In particular, discussion regarding softmax response and score-based vs abstaining classifiers would be added.
> We would add this either to the main text (Sec 2\) with the additional page / space of the camera ready (alongside some reduction on the GVI / DRO section) or via an additional appendix section.
>
> > Empirical evaluation can be improved (datasets)
>
> Thanks for the dataset recommendation. Please see the shared rebuttal response section. We have added some preliminary results utilizing the MedMNIST collection.
>
> > Q1: Softmax Response (SR)
>
> Our work primarily started from examining rejection approaches which were inspired by surrogate loss functions and cost-sensitive losses. These approaches formed the baselines which we considered (many of these papers did not consider softmax response).
> Although we do not include softmax response as a baseline, we note that the density-ratio rejectors have a strong connection to this method.
> In the case of the KL-divergence rejectors with the practical trick employed (as noted in Line 271 onwards), our rejector will threshold the model’s predictive entropy. This is similar to the maximum softmax score.
> The theoretical framework of prediction using density-ratios presents a generalization of this elucidating further connections to DRO / GVI. We further believe that the sample complexity bounds do not have an equivalent in SR.
>
> Empirically, as a result, our approach and naive threshold rejection using the softmax scores should be comparable (when both utilize temperature scaling).
> The component of SR that we are missing is the calculation of the threshold according to a desired coverage guarantee. This would be a nice direction for future work to automatically prescribe $\\tau$.
>
> We will include this discussion and connection to the SR in the next version of the paper.
>
> > Q2: DEFER without expert
>
> One can utilize DEFER without human predictions by utilizing a ground-truth expert (one returning the data’s label). In particular, it has been shown that this interpretation provides a surrogate loss function of the 0-1-c loss function. This has been shown in \[a, see Appendix A.2\] and \[b\] which generalizes this. We are happy to include this discussion to make this baseline choice clear.
>
> \[a\] Charoenphakdee, Nontawat, et al. "Classification with rejection based on cost-sensitive classification." International Conference on Machine Learning. PMLR, 2021\.
>
> \[b\] Cao Y, Cai T, Feng L, et al. Generalizing consistent multi-class classification with rejection to be compatible with arbitrary losses. NeurIPS, 2022\.

---

> > ### Comment · Reviewer_dzAs · 2024-08-08
> > **Response to the Rebuttal**
> >
> > I thank the authors for their rebuttal, and I hope they found the references to the MedMNIST collection helpful.
> > I would like to add a small comment on the following:
> >
> > > The component of SR that we are missing is the calculation of the threshold according to a desired coverage guarantee. This would be a nice direction for future work to automatically prescribe $\tau$.
> >
> > Some approaches, as noted by the authors in the response to Reviewer MrTu, add a coverage constraint to the loss that is minimized. However, empirical evidence does not support using these models vs. SoftMax Response (see again e.g. [a,b]).
> > In any case, given a target coverage $c$, both approaches require a calibration set and estimate the threshold $\tau$, considering the $(1-c)$ quantile of the confidence on the calibration set.
> > Another option is resorting to cross-fitting to estimate the $\tau$ without the extra calibration set but exploiting test folds and stacking confidences (see, e.g., [c]).
> >
> > To conclude, I think this paper should be considered for acceptance.
> >
> > [a] - Feng, Leo, Mohamed Osama Ahmed, Hossein Hajimirsadeghi, and Amir H. Abdi. "Towards Better Selective Classification." In The Eleventh International Conference on Learning Representations.
> > [b] - Pugnana, Andrea, Lorenzo Perini, Jesse Davis, and Salvatore Ruggieri. "Deep neural network benchmarks for selective classification." arXiv preprint arXiv:2401.12708 (2024).
> > [c] - Pugnana, Andrea, and Salvatore Ruggieri. "A model-agnostic heuristics for selective classification." In Proceedings of the AAAI Conference on Artificial Intelligence, vol. 37, no. 8, pp. 9461-9469. 2023.

---

> > > ### Author Response · Authors · 2024-08-13
> > >
> > > We thank the reviewer for their support of the paper and for providing some additional context on the empirical weakness of the other baselines. We briefly considered cross-fitting to find thresholds values similar to [c], however did not go that route due to increases in computational costs for larger datasets. Nevertheless, we will include this discussion in to next revision and also add citations for previous work which have successful utilized this approach.

---

### Official Review · Reviewer_MrTu · 2024-07-09

**Soundness:** 3
**Presentation:** 3
**Contribution:** 3
**Rating:** 5
**Confidence:** 4

**Summary:**

Classification with rejection emerges as a learning paradigm that allows models to abstain from making predictions. Traditional rejection learning methods typically modify the loss function, enabling models to explicitly reject making predictions when they are inaccurate or uncertain. These methods rely on providing good class probability estimates and often require training models from scratch. This paper proposes an interesting perspective to reconsider rejection not from the standpoint of the loss function, but rather from the perspective of distributions. By optimizing the risk of the loss with a $\varphi$-divergence regularization term, it seeks an idealized data distribution that maximizes the performance of pretrained models.

**Strengths:**

1. This paper takes a more interesting approach to rejection compared to previous papers.
2. In theory, approaching from the perspective of distributions holds better prospects compared to considering single probabilities alone.
3. The theoretical richness of this paper provides strong evidence for its claims.

**Weaknesses:**

1. For high-dimensional data, estimating density ratios and computing idealized distributions can be very challenging, implying that this method may only address low-dimensional data and its application scenarios are limited.
2. This method involves a large number of hyperparameters: $\lambda$, $\tau$, $\alpha$, $T$, which pose a significant burden for optimization.
3. The experiments in the paper may only be conducted on specific datasets and under certain conditions (some simple datasets), which could limit their generalizability to broader scenarios.

**Questions:**

1. Although theoretically estimating the distribution $P(y|x)$ holds more promise than estimating a single probability $\max P(y|x)$, in practical situations, estimating the distribution seems less feasible compared to single probability estimates, especially given the various calibration methods available for single probabilities.
2. How did you design the expert in the experiments where you compared methods [1] from Learning to Defer？
3. Since comparing with [1], it seems more reasonable to compare with [2], which is more relevant to the focus of this study.

[1] Hussein Mozannar and David Sontag. Consistent estimators for learning to defer to an expert. ICML 2020.

[2]  Cao Y, Cai T, Feng L, et al. Generalizing consistent multi-class classification with rejection to be compatible with arbitrary losses. NeurIPS, 2022.

**Limitations:**

N/A.

---

> ### Author Rebuttal · Authors · 2024-08-07
>
> We thank the Reviewer for praising the interesting nature of our proposed approach and for the insightful questions. We are glad that the Reviewer appreciates the different approach we have taken and hope that the connections to DRO / GVI / distributions can lead to new theories. Please find the per-point rebuttal below.
>
> > \[...\] this method may only address low-dimensional data \[...\]
>
> We agree that with a naive application of this approach, there may be problems in high dimensions. This is a weakness shared by other density ratio-style algorithms like importance weighting.
> We would however note that there has been success in leveraging representation learning to operate in a lower dimension. This can be used as a potential work around, especially if the model weights of the original classifier is available.
>
> > This is a weakness seen in other density ratio-style approaches like importance weighting.
>
> We would like to comment on the hyperparameters noted by the reviewer.
>
> - We are unsure what hyperparameter $T$ the reviewer is referring to.
> - $\\tau$ for the threshold can be treated as a hyperparameter. But it can also be used as a value to be found to satisfy a specific risk-coverage requirement. This can be equivalent to $c$ considered in the baseline approaches.
>   Actually, **Reviewer dzAs** reminded us of some work which picks thresholds according to a coverage-modified-risk requirement \[a\]. As future work, this might be an interesting future direction to our work.
> - We agree that tuning may present some challenges $\\lambda$ and $\\alpha$. However, we note that picking $\\lambda$ and $\\alpha$ is equivalent to picking an appropriate divergence. This is equivalent to how picking a surrogate loss for the classifier-rejection 0/1 loss function has many options.
>
> One additional note that we make about hyperparameter tuning of our approach, is that although there are potentially more individual parameters one may want to pick from, it is substantially cheaper to tune than the other baselines. Indeed, the process of learning the 1D normalization constant $b$ required for density ratio rejectors (Line 287\) is substantially cheaper than learning typically a whole network as required by the baselines. For our approach, we can keep the base classifier fixed and we only need to learn $b$ (see Appendix P.III for discussion on the number of tunable parameters of the approaches).
>
> \[a\] Geifman, Yonatan, and Ran El-Yaniv. "Selective classification for deep neural networks." Advances in neural information processing systems 30 (2017).
>
> > Experiments (Weakness 3\)
>
> Please see the shared rebuttal section. Other reviewers also commented about adding other datasets for strong evaluation. We have taken **Reviewer dzAs**’s recommendation of considering datasets in the MedMNIST collection.
>
> > Q1: Calibration
>
> Although calibration for the multiclass case is more challenging in practice than perhaps estimating $\\max P(y \\mid x)$, we shouldn’t dismiss theoretically motivated frameworks due to this reason.
> From a practical point of view, there are still many approaches which have shown reasonable success. For example, the standard temperature scaling approach that we utilize in the paper; or various different ensembling-style methods which we do not explore.
>
> > Q2: Experts for DEFER
>
> In essence, we are assuming that the experts replicate the ground truth labels.
> This is equivalent to the reduction in \[b, see Appendix A.2\] which provides a surrogate loss function for rejection. Essentially, the learning to DEFER loss function can be used to learn a general $g \\colon \\mathbb{R}^d \\rightarrow \\mathbb{R}^{K+1}$ classifier for $K$ classes with the last $K+1$ output dimension being used for rejection.
> This reduction / interpretation was also commented by Cao et al. (reference \[2\] of the reviewer) as their method generalizes this instantiation of DEFER.
>
> We will make sure to clarify how we are using DEFER in the next version of the paper.
>
> \[b\] Charoenphakdee, Nontawat, et al. "Classification with rejection based on cost-sensitive classification." International Conference on Machine Learning. PMLR, 2021\.
>
> > Q3: Comparing with \[2\]
>
> Apologies, this seems to be a clarity issue. The GCE baseline tested in the paper is actually \[2\] (in retrospect, this may not be as clear as we also cited the general cross-entropy loss function’s original paper). Following up on Q2 as well, this GCE baseline can be interpreted as a generalization of the surrogate loss derived from DEFER.

---

> > ### Comment · Reviewer_MrTu · 2024-08-11
> > **Response**
> >
> > Thank you for your elaborated response to my review.
> >
> > I understand that the author's work is interesting and meaningful, but I still maintain my concerns about its validity on high-dimensional datasets (despite the addition of experiments using MedMNIST). As we are now gradually entering the era of large models, could you provide some more experiments on high-dimensional data, such as CIFAR10, regardless of the results? I hope this will help to better understand the limitations of the method.

---

> > > ### Author Response · Authors · 2024-08-13
> > >
> > > We thank the reviewer for their additional comment. We are happy to include results on CIFAR10. In the following, we have mirrored the setting of the additional MedMNIST / OrganSMNIST experiments presented in our global response.
> > > Of these settings, we note that this uses a Resnet18 base model for the classifier and baselines; and that the specific hyperparameter grid for $c$ is reduced to ensure that the experiments completed in time for the discussion deadline. Random cropping and horizontal flipping data augmentation are utilized in training for all approaches.
> > >
> > > We present an additional 85% coverage table as the coarseness of the $c$ grid provided better comparison for some of the baselines.
> > >
> > > ### CIFAR-10 (80% Coverage)
> > >
> > > |    | Model Name            | Hyperparameter   | Accuracy       | Coverage        |
> > > |---:|:----------------------|:-----------------|:---------------|:----------------|
> > > |  0 | Base Clf              | -                | 90.095 (0.244) | 100.000 (0.000) |
> > > |  1 | $\rm KL-Rej$          | 0.74             | 97.285 (0.222) | 80.165 (0.428)  |
> > > |  2 | $(\alpha=3)\rm{-Rej}$ | 0.74             | 97.236 (0.212) | 80.433 (0.350)  |
> > > |  3 | $\rm PredRej$         | 0.1              | 90.271 (0.415) | 94.792 (6.079)  |
> > > |  4 | $\rm CSS$             | 0.15             | 95.610 (0.274) | 84.077 (0.616)  |
> > > |  5 | $\rm DEFER$           | 0.2              | 93.639 (0.964) | 82.530 (3.919)  |
> > > |  6 | $\rm GCE$             | 0.25             | 93.807 (0.738) | 82.710 (1.115)  |
> > >
> > >
> > > ### CIFAR-10 (85% Coverage)
> > >
> > > |    | Model Name            | Hyperparameter   | Accuracy       | Coverage        |
> > > |---:|:----------------------|:-----------------|:---------------|:----------------|
> > > |  0 | Base Clf              | -                | 90.095 (0.244) | 100.000 (0.000) |
> > > |  1 | $\rm KL-Rej$          | 0.62             | 95.966 (0.283) | 85.313 (0.490)  |
> > > |  2 | $(\alpha=3)\rm{-Rej}$ | 0.58             | 96.059 (0.305) | 85.077 (0.469)  |
> > > |  3 | $\rm PredRej$         | 0.1              | 90.271 (0.415) | 94.792 (6.079)  |
> > > |  4 | $\rm CSS$             | 0.2              | 94.339 (0.158) | 87.783 (0.251)  |
> > > |  5 | $\rm DEFER$           | 0.25             | 93.423 (0.698) | 87.837 (1.206)  |
> > > |  6 | $\rm GCE$             | 0.3              | 93.295 (0.781) | 85.720 (0.682)  |
> > >
> > >
> > > ### CIFAR-10 (90% Coverage)
> > >
> > > |	| Model Name        	| Hyperparameter   | Accuracy   	| Coverage    	|
> > > |---:|:----------------------|:-----------------|:---------------|:----------------|
> > > |  0 | Base Clf          	| -            	| 90.095 (0.244) | 100.000 (0.000) |
> > > |  1 | $\rm KL-Rej$      	| 0.52         	| 94.314 (0.240) | 90.070 (0.445)  |
> > > |  2 | $(\alpha=3)\rm{-Rej}$ | 0.38         	| 94.179 (0.264) | 90.420 (0.406)  |
> > > |  3 | $\rm PredRej$     	| 0.1          	| 90.271 (0.415) | 94.792 (6.079)  |
> > > |  4 | $\rm CSS$         	| 0.25         	| 93.134 (0.641) | 90.025 (0.705)  |
> > > |  5 | $\rm DEFER$       	| 0.3          	| 93.656 (0.393) | 88.352 (2.164)  |
> > > |  6 | $\rm GCE$         	| 0.3          	| 93.295 (0.781) | 85.720 (0.682)  |
> > >
> > > ### Discussion
> > >
> > > We make a few comments on the reported results.
> > >
> > > - For this higher dimensional dataset, the same patterns emerge when comparing to MedMNIST and the paper’s original datasets in the noiseless case. Our approach can perform better than the baselines.
> > > - An additional observation on the best $c$ taken for the baseline approaches: the optimal cost of rejection per coverage target depends greatly on the approach with no consistency across all baselines. This demonstrates that tuning of $c$ can be difficult in practice (especially as each $c$ requires an entire model to be trained).
> > > - The increase in dimension from OrganSMNIST/MNIST (1x28x28) to CIFAR-10 (3x32x32) does not seem significant enough to cause any practical issues.
> > >
> > > Additional to these direct comments, we would also like to make some other notes. Although the general framework we propose requires the learning of density ratios to determine the rejector, the closed-form solutions and practical trick (Line 271\) that we utilize does not require explicit learning of a density ratio. Instead, transforming a base model’s calibrated output is exploited (with only the normalization constant $b$ or $Z$ requiring approximation). As such, issues of high dimensional data for the current implementation would mainly be linked to difficulties in calibration in high dimension (which is a valid concern).
> > >
> > > Nevertheless, we hope these additional results address the reviewer’s concerns for at least this CV dataset. We will of course make a note of this limitation and the corresponding discussion in the next revision. This would be particularly useful in future work where different divergences or modifications do require an explicit distribution or density-ratio to be learned.

---

> > > > ### Comment · Reviewer_MrTu · 2024-08-14
> > > >
> > > > Thank you to the authors for providing the additional experiments. I believe that after including these extra experiments, this paper is quite good, and I would be happy to raise my score to 5.

---

### Official Review · Reviewer_MiAk · 2024-07-12

**Soundness:** 3
**Presentation:** 2
**Contribution:** 3
**Rating:** 6
**Confidence:** 3

**Summary:**

This paper proposes a novel method for classification with rejection based on density ratio estimation. Density ratio is estimated between the data distribution P and the "idealized" distribution Q, where Q is a distribution such that the model can have good prediction performance, while this distribution is still similar to the original data distribution P. The objective function is proposed to formulated as a risk minimization with a $\phi$-divergence regularization. Theoretical analysis is also provided to justify the soundness of the proposed method. The experiment shows that the proposed method is effective compared with well-known existing approaches.

**Strengths:**

1. Proposed method is theoretically guaranteed and can support multiclass classification case. To me, it is not that straightforward to use density ratio estimation approach for multiclass classification with rejection and I appreciate this strength of the paper.
2. The proposed method and formulation discussed in the paper is quite general, and the discussion for Regression case also provided in appendix, suggesting the generality of the proposed method. Divergence that studies in this paper is not only KL-divergence, but different divergences were also studied.

**Weaknesses:**

1. Unfortunately, I found the experiment section writing and result is not quite well-written compared with other sections. Perhaps there is a better way to analyse and understand practical behavior of the proposed method.
 - The figure shows in the paper does not really show the superiority of the method except in HAR (clean) and Gas Drift (clean). It is also quite hard to see since the starting point of each method is different. Perhaps table representation or different way to show the result might help.
  - Only three simple datasets were used in this paper and the discussion using these datasets still look difficult to conclude something as several methods are competitive.
  - In my understanding, KL and $\alpha$ divergence give almost exactly the same performance in all cases. Is this really the case?
  The study of hyperparameter sensitivity can also be beneficial to practitioners
  - Analysis of what kind of dataset characteristic makes the proposed distributional method more effective than the traditional method would be very useful. I feel existing methods such as CSS or the classifier-rejector approach might feel more like a direct approach to solving classification with rejection. Nevertheless, it is great to explore other approaches.
  -  As a result, I believe revising the experiment section can significantly improve the paper.
2. I found the writing and organization of this paper can be significantly improved. It takes until page 4 to state the proposed method, leaving only a small space for the experiment section. Perhaps some discussion of theory for $\alpha-divergence and background of (DVI, DRO) can be wrapped up more compactly and defer to the appendix to make the paper more self-contained and provide sufficient discussion on experiments.
  - For reading experience, it might also be useful to highlight the final objective function (or use a latex Algorithm environment) to outline what exactly we need to do to use the proposed method so that practitioners can quickly understand how the method works. I found section 4 is quite difficult to follow. In this form, we have to go through the whole paper and mix all the ingredients to get an idea how to implement the proposed method (e.g., Normalization, how to get a(x), inverse function of $\varphi$.

**Questions:**

1. Is there any superiority of $\alpha=3$ divergence over KL-divergence? I might have missed it.
2. Is it written in the main body of the paper what $\varphi'$ is (Eq. 11)? I believe it is a derivative of $\varphi$ over X? I am sorry if I missed it. If it is not written it should be so because for example, "'" can mean different things like how L' is used in this paper.
3. It seems the proposed method is quite weak under noisy data. Is there any possible explanation why this is the case?
4. The Theorem 4.2 (informal)  states that there exists a threshold to achieve Chow's rule if original $h$ is correct looks quite restrictive to me in the sense that if h is really optimal, then we can just use h to achieve Chow's rule and there is no need to consider density ratio estimation. Can we say anything when h is not optimal?


Minor comment:
I think it is more common to call "classification-rejection" approach, a classifier-rejector approach (like Ni et al., 2019 [44]), or predictor-rejector approach (Mao et al., 2024) [39].

**Limitations:**

Limitations about the need to estimate P(Y|X) with model h were discussed, suggesting that this approach requires estimating class-posterior probability.

(Minor) broader impacts were discussed in Appendix L and perhaps it is better to put them in the main body if possible.

Overall, I found the discussion about limitations is appropriate.

---

> ### Author Rebuttal · Authors · 2024-08-07
>
> We thank the reviewer for their detailed reading and many suggestions for improving the readability of the paper. We will change “classification-rejection” to “classifier-rejector” as per the reviewer’s suggestion and space-permitting, will include the broader impact section in the main-body (or add a compact part of it in the conclusion). Please find other rebuttal points below.
>
> > Perhaps table representation or different way to show the result might help. \[...\] Only three simple datasets were used in this paper
>
> Thank you for the suggestion. We will add tables corresponding to different coverage targets to the paper. Please see the shared section to see a demonstration on an additional dataset added in the rebuttal period. We hope that the added dataset from the MedMNIST collection also alleviates the concern that the datasets we consider are only simple.
>
> > KL and $\\alpha$ divergence give almost exactly the same performance in all cases \[...\] \+ Q1
>
> We would first like to note that we do include a hyperparameter sensitivity plot in Figure VII of the appendix. This tests different $\\alpha$s greater than 1\.
> In terms of ranges of $(\\alpha \\geq 1)$-divergences, the change in performance-to-coverage is minor. This can also be further seen in the table provided in the shared response section (which more clearly shows that although the results are not identical, they are not significantly different).
> One difference we did find was that the coverage curve “shortens” for $\\tau \\in \[0, 1\]$. Although $\\tau$ can be increased in practice, the cut-off in coverage for $\\tau \= 0$ shortens.
>
> > Analysis of what kind of dataset characteristic makes the proposed distributional method more effective \[...\] \+ Q4: Weakness under noise
>
> We believe that the dataset characteristic and your question about weakness under noise are linked. In general, in the noiseless case, there doesn’t seem to be a specific characteristic between what makes our proposed distributional rejection more effective. In general, it seems that the distributional rejection is slightly better in this clean case (results in tabular format, see general response).
> However, it becomes clear that from the drop in performance in the noisy dataset setting, that distributional rejection using $(\\alpha \\geq 1)$-divergences with our current proposed implementation is not as robust to noise as other methods. This makes sense as the current approach relies on the calibration of models, which would break under distribution shift.
> A note here is that future work examining divergences used in robustness may help alleviate this issue.
>
> > \[...\] writing and organization of this paper can be significantly improved.
>
> We thank the reviewer for their suggestions on improving the writing and organization of the paper. The following summarizes the small changes regarding this topic that will be made:
>
> - Additional space provided by the camera ready will be utilized for further experimental discussion (including some of the above discussed points in this rebuttal section) and a table summarizing experiments.
>   A related work (sub)section dedicated for rejection will be added as suggested by **Reviewer dzAS** (possibly deferred to the appendix).
>   The content for DRO and GVI will be reduced if needed.
> - To shortcut the theoretical parts of the paper, we will add an pseudo-code block next to Figure 1 which summarizes the abstract algorithm of distributional rejection. In addition, we will reference specific equations used to define the ratio $\\rho$.
>   Additional text referencing this will also be added in the intro and start of Sec 3\.
> - Frame / highlight environments will also be used on equations referenced in the pseudo-code, as suggested by the reviewer.
> - We believe that the current structure of the Sec 2-4 works well from a theoretical analysis point of view. And the proposed changes above will shortcut and identify key components for implementation.
>
> We are confident that with the \+1 page, adding additional discussion, tables, and cosmetic highlights can be complete to improve the paper’s presentation.
>
> > Q2: Clarity on $\\phi'$ and $L'$
>
> We would like to confirm that $\\phi'$ is the derivative. Actually, the $L'$ notation also can be interpreted as a functional derivative of $L$ on $Q(x)$. We will mention this notational choice to clarify why notation is set this way. Do feel free to comment if this still makes the notation of $L’$ confusing. We are open to changing this to another symbol if required, eg, via font $\\mathrm{L}$ or with a bar $\\bar{L}$.
>
> > Q4: Theorem 4.2
>
> We agree with the reviewers comment that the theorem is restrictive. Its purpose is just to verify that the proposed distributional rejection method produces Chow’s rule under theoretically optimal conditions.
> When the optimal classifier is not optimal, the optimal rejection becomes a thresholding function over the pointwise risk of the model (in the language of CPE classification, see reference \[48, 49\]).

---

> > ### Comment · Reviewer_MiAk · 2024-08-12
> > **Thank you for your feedback**
> >
> > Thank you for the author's rebuttal. I also have read other reviews.
> >
> > Given the fact that the organization will be modified as mentioned in the rebuttal and additional experiments provided, I would like to raise the score to 6 (Weak accept). I am aware of the concern of the high-dimensionality problem suggested by other reviewers. I believe the approach taken by this paper is innovative and it is also interesting to design other algorithms that follow this principle of identifying idealized distribution to use it as a criterion for rejection, which could be preferable to the current method in the high dimensionality regime.

---

> > > ### Author Response · Authors · 2024-08-13
> > >
> > > We thank the reviewer for raising their score. We are also interested in seeing new algorithms which exploit our proposed perspective of rejection. On the topic of high-dimensional data, the reviewer may be interested in the CIFAR-10 setting proposed by Reviewer MrTu. We have presented experimental results mirroring those provided for MedMNIST / OrganSMNIST and have reached similar conclusions to those expressed in the original global discussion.

---

### Official Review · Reviewer_m8bi · 2024-07-15

**Soundness:** 3
**Presentation:** 2
**Contribution:** 3
**Rating:** 6
**Confidence:** 3

**Summary:**

This paper propose a new classification algorithm with abstention by learning a density ratio between an "idealized" distribution to the data distribution: if the density ratio is small, the classifier rejects to predict.
The proposed learning framework is general, and is a function of the choice of $f$-divergence.
The paper studies two concrete choices of divergences: KL- and $\a$-divergences, which admit a closed form rejector.
The paper provides some experimental supports for the proposed ideas with neural-net classifiers.

**Strengths:**

The paper studies an important problem of classification with abstention.
The proposed mathematical framework is quite neat and has nice connections with GVI and DRO.
The math is quite crisp throughout the paper.

**Weaknesses:**

While the proposed method sounds reasonable, I have some concerns/questions in the presentation.
Please see questions and suggestions below.

**Questions:**

**Questions**
- What is the meaning of "The maximization is adversarial over the marginal label distribution" in line 128?
- I had hard time to understanding the meaning of the "idealized rejection distribution" (as called. in line 157), especially because the following sentence was extremely confusing:
```
Given Definition 3.1 for idealized distribution, small values of $\rho(x)$ corresponds to regions of the input space $\mathcal{X}$ where having lower data probability would decrease the expected risk of the model.
```
I can't follow this sentence. Can you please elaborate this?
- Further, I can't understand why does it make sense not to reject where $P(x)$ is relatively small? If the likelihood of occurrence is small, shouldn't we also avoid prediction at that point as the classification will be likely wrong? Or is the argument that we do not need to care the small probability mass in the first place as it won't incur a significant loss in expectation?
- In the experiments, the proposed algorithms seem to perform well in the high-coverage regime, while dominated by existing methods in the low-coverage regime. Do you have any explanation on this?

**Suggestions**
- Compared to the quality of math, there are way too many typos and grammatical errors. To spot a very few:
  - line 23: "While" some approaches avoid...
  - line 25: ... approach aim**s**...
  - line 74: ... an addition**al**...
There are too many of these throughout, so please revise them carefully.
- Please consider moving lines 84-85 after line 76.
- Please put subsections in Section 4 for readability.

**Limitations:**

Limitations of the framework are noted by the authors.

---

> ### Author Rebuttal · Authors · 2024-08-07
>
> We thank the reviewer for their praise of the mathematical presentation of the paper and their careful reading. We are happy to change the paper as per the reviewer's suggestions and will ensure that the typos and grammar improves. In what follows, we will answer **Reviewer m8bi** additional questions.
>
> > What is the meaning of "The maximization is adversarial over the marginal label distribution" in line 128?
>
> This line was to clarify that in DRO for label noise (specifically reference \[60\]), typically the distribution which is “adversarial” only applies to the marginal label distribution and not the full joint distribution over both inputs and labels. Thus the set of adversarial joint distributions $Q(y) P(x | y)$ are generated by varying the marginal over labels $Q(y)$ whilst keeping the conditional equal to the data $P(x | y)$.
>
> > I had hard time to understanding the meaning of the "idealized rejection distribution" (as called. in line 157\)
>
> The best intuitive way of thinking of the idealized distribution definition is to remove the regularization term in Definition 3.1. This corresponds to the distribution which minimizes the current model’s loss. Hence its “ideal” for the current model / set of parameters.
>
> > especially because the following sentence \[…\]
>
> For the follow up sentence, we further elaborate.
> When $\\rho(x)$ is small, this implies that $Q(x) \\ll P(x) \\leq 1$. (Recalling that $Q$ is the idealized distribution and $P$ is the data distribution.)
> Further note that from Definition 3.1 / Equation (8), $Q(x)$ being small implies that the loss $L’(x)$ cannot be too large (as otherwise we should reallocate mass from other x’s).
> Hence wrt the data distribution $P(x)$, moving mass away for regions of $x$ where $\\rho(x)$ is small reduces the loss function.
>
> > Further, I can't understand why does it make sense not to reject where is relatively small? \[...\] Or is the argument that we do not need to care the small probability mass in the first place as it won't incur a significant loss in expectation?
>
> The reviewers comment about the values corresponding to loss probability mass and thus no significant loss penalty in expectation is correct. Our rejection approach focuses on losses at different inputs values, as per Definition 3.1.
>
> Of course, this is from the perspective of our framework, so practically, various rejection rules may be needed to account for different types of uncertainty.
> A future direction may be to consider alternatives to expected loss which account for low $P(x)$. For example CVaR etc.
>
> An alternative theoretical reason for this style of thresholding choice is Theorem 4.2 in the paper. With our specific choice of thresholding / rejection on the ratio $\\rho(x) \= Q(x) / P(x)$, we recover Chow’s rule. The alternative of rejecting based on $P(x)$ would not yield this equivalence.
>
> > In the experiments, the proposed algorithms seem to perform well in the high-coverage regime, while dominated by existing methods in the low-coverage regime. Do you have any explanation on this?
>
> Good question. Firstly, we note that the trade-off curves can be extended further by increasing $\\tau > 1$. We suspect the difference in performance to be related to the quality of calibration. If the model was perfectly calibrated, then the density approach would be optimally rejecting. As such, deviation / suboptimality would be due to calibration error.
> This explanation is partially supported by the drop in performance of our approach on noisy variants of the dataset. Here the training set (and data used for calibration) is not representative of the test set which would result in a model miss-calibrated on the test set.
>
> For two additional notes:
>
> We also point the reviewer to the results of the MedMNIST dataset \-- additionally added in the rebuttal \-- in the shared response section. Here we find that for the lower 80% regime, our approach can perform better than the baselines for a more complicated dataset.
>
> A reason why the baselines do not provide / perform well in the extremely high-coverage regime (\~ 95%) is that it is difficult to tune the rejection costs $c$ hyperparameter to allow for fine tuning of the coverage. As the rejection decision of these baseline involve training a whole other model, the cost of tuning requires training several NNs.

---

> > ### Comment · Reviewer_m8bi · 2024-08-12
> >
> > I appreciate the authors' rebuttal.
> > Please carefully incorporate the answers in the revision.
> > I believe that it is important to elaborate the reasoning behind using the idealized distribution.
> > I will keep the score as is.

---

> > > ### Author Response · Authors · 2024-08-13
> > >
> > > We thank the reviewer for their response. We will ensure we incorporate the provided feedback and response into the next revision.

---

### Author Rebuttal · Authors · 2024-08-07

We thank the reviewers for their thoughtful comments and various suggestions.
We are grateful to the reviewers for recognizing the novelty of our work, providing a “\[...\] more interesting approach to rejection compared to previous papers” (**Reviewer MrTu**). We appreciate the comments that our work has “\[...\] clear contributions advancing the state of the art in abstaining classifiers.” (**Reviewer dzAs**) including “\[...\] support multiclass classification case \[... which\] is not that straightforward \[...\] (**Reviewer MiAk**); and the recognition that our approach is “\[...\] quite neat and has nice connections with GVI and DRO \[... and the\] math is quite crisp throughout the paper.” (**Reviewer m8bi**).

Please find the per point rebuttal below. In this shared section, we additionally provide results using the MedMNIST dataset collection and provide some samples for presenting the performance of the results via tables.

## Additional datasets (MedMNIST) & Tables

**Reviewer MiAk, MrTu and dzAs** mentioned that a potential weakness of the current paper is the simplicity of the currently tested datasets. Noise was utilized to add complexity to the dataset settings and to provide cases where the base classifier does not reach a high (near 100%) accuracy. To supplement the experiments on these datasets and their noisy label noise variants, in the rebuttal we provide some additional experimental results on the MedMNIST dataset collection recommended by **Reviewer dzAs**.

Additionally, **Reviewer MiAk** mentioned that presenting results in tabular format might enhance the readability of the experimental results settings. In this rebuttal section, we present accuracy tables for the MedMNIST dataset results, where we select hyperparameters (and their coverage) which are closest to a specific coverage goal. In particular, we will look at 90% coverage (high coverage) and 80% coverage (lower coverage) regimes.
We have also included preliminary tables in the attached PDF for Figure 2 of the paper, where the coverage target is 80%.

In the following, we present results for the OrganSMNIST dataset from the MedMNIST dataset collection. This is a 2D 28x28 classification task over grayscale images and 11 classes. We note that for this dataset, the base classifier (which we use a ResNet18 following the MedMNIST paper \[a\]) is not close to 95-100%.
The general setup is identical to that in the paper (5-fold cross validation \+ grid search over hyperparameters of either $\\tau$ or $c$). The only difference is that there is a slightly reduced hyperparameter search of $c$ for ranges between 0.01 and 0.3 (rather than 0.01 to 0.5; DEFER had some additional hyperparameters tested) to complete these preliminary experiments on time for the rebuttal.

The result tables are given as follows:

### OrganSMNIST (80% Coverage Target)

|	| Model Name        	| Hyperparameter   | Accuracy   	| Coverage    	|
|---:|:----------------------|:-----------------|:---------------|:----------------|
|  0 | Base Clf          	| -            	| 90.119 (0.867) | 100.000 (0.000) |
|  1 | $\rm KL-Rej$      	| 0.64         	| 97.548 (0.359) | 80.512 (0.910)  |
|  2 | $(\alpha=3)\rm{-Rej}$ | 0.6          	| 97.660 (0.400) | 80.144 (0.795)  |
|  3 | $\rm PredRej$     	| 0.2          	| 89.938 (4.094) | 81.662 (13.320) |
|  4 | $\rm CSS$         	| 0.08         	| 96.566 (0.854) | 81.306 (1.255)  |
|  5 | $\rm DEFER$       	| 0.14         	| 94.068 (0.677) | 80.743 (3.946)  |
|  6 | $\rm GCE$         	| 0.1          	| 93.973 (0.850) | 80.068 (2.677)  |

### OrganSMNIST (90% Coverage Target)

|	| Model Name        	| Hyperparameter   | Accuracy   	| Coverage    	|
|---:|:----------------------|:-----------------|:---------------|:----------------|
|  0 | Base Clf          	| -            	| 90.119 (0.867) | 100.000 (0.000) |
|  1 | $\rm KL-Rej$      	| 0.54         	| 93.757 (0.308) | 90.000 (1.241)  |
|  2 | $(\alpha=3)\rm{-Rej}$ | 0.4          	| 93.565 (0.327) | 90.639 (1.088)  |
|  3 | $\rm PredRej$     	| 0.22         	| 91.556 (1.603) | 90.298 (6.066)  |
|  4 | $\rm CSS$         	| 0.19         	| 92.437 (0.313) | 90.361 (1.619)  |
|  5 | $\rm DEFER$       	| 0.41         	| 91.557 (0.907) | 91.916 (1.496)  |
|  6 | $\rm GCE$         	| 0.25         	| 92.203 (0.346) | 90.714 (1.429)  |

### Discussion

From these results, interestingly our density-ratio rejectors are actually superior over these datasets. Thus, our method has practical relevance to datasets beyond those previously tested.

Some other observations include:

- There are minor differences between KL and $\\chi^2$ / $\\alpha \= 3$. However, the differences are not significant.
- Performance of the density-ratio rejectors are not tied to near perfect accuracy of the base classifier.
- Our method can perform better at high and lower coverage target regimes.

We will add the new experimental result in the next version of the paper alongside other 2D datasets in the MedMNIST collection.

\[a\] Yang, Jiancheng, et al. "Medmnist v2-a large-scale lightweight benchmark for 2d and 3d biomedical image classification." Scientific Data 10.1 (2023): 41\.

---

### Decision · Program_Chairs · 2024-09-25

**Decision:**

Accept (poster)

**Comment:**

In this work, the authors develop a new algorithm for classification with rejection by learning an idealized distribution using the model, and rejecting a classification if the ratio of the idealized density to the data density is too large. Efficacy is demonstated using both theoretical error bounds and a short series of experiments. Reviews were generally positive, but criticised the limited number of experiments and acknowledged that both density estimation and the ratio estimator are likely to succumb to the curse of dimensionality for larger problems. In their rebuttal, the authors provided a series of additional experiments highlighting potential efficacy for problems of larger size. After the discussion phase, all reviewers agreed the paper was of high quality with the additional results. Subject to these inclusions, I recommend acceptance.